# ARE TRANSFORMERS UNIVERSAL APPROXIMATORS OF SEQUENCE-TO-SEQUENCE FUNCTIONS?

**Chulhee Yun**[*]
MIT
chulheey@mit.edu

**Srinadh Bhojanapalli**
Google Research NY
bsrinadh@google.com

**Ankit Singh Rawat**
Google Research NY
ankitsrawat@google.com

**Sashank J. Reddi**
Google Research NY
sashank@google.com

**Sanjiv Kumar**
Google Research NY
sanjivk@google.com

## ABSTRACT

Despite the widespread adoption of Transformer models for NLP tasks, the expressive power of these models is not well-understood. In this paper, we establish that Transformer models are universal approximators of continuous *permutation equivariant* sequence-to-sequence functions with compact support, which is quite surprising given the amount of shared parameters in these models. Furthermore, using positional encodings, we circumvent the restriction of permutation equivariance, and show that Transformer models can universally approximate *arbitrary* continuous sequence-to-sequence functions on a compact domain. Interestingly, our proof techniques clearly highlight the different roles of the self-attention and the feed-forward layers in Transformers. In particular, we prove that fixed width self-attention layers can compute *contextual mappings* of the input sequences, playing a key role in the universal approximation property of Transformers. Based on this insight from our analysis, we consider other simpler alternatives to self-attention layers and empirically evaluate them.

## 1 INTRODUCTION

Self-attention based Transformer networks (Vaswani et al., 2017) have been at the center of the recent progress on various natural language processing (NLP) tasks, including machine translation (Vaswani et al., 2017), language modeling (Radford et al., 2018; 2019), and question answering (Devlin et al., 2018; Yang et al., 2019; Liu et al., 2019). All these tasks involve learning models that map an input sequence of tokens to an output sequence of tokens. Transformers make it feasible to train large models to approximate these sequence-to-sequence functions due to their ability to process the input tokens in a parallel way, as opposed to the sequential nature of RNNs and LSTMs.

A Transformer block consists of two kinds of layers: a self-attention layer and a token-wise feed-forward layer, with skip connections present in both layers. The self-attention layer transforms each input token embedding using a weighted combination of the embeddings of all tokens in the input sequence, where weights are generated by pairwise dot-products among the input token embeddings. The token-wise feed-forward layer then independently processes each of these modified input token embeddings without any interaction among them. Notably, Transformers employ parameter reuse across tokens, as both layers use the same parameters to process each token. Moreover, Transformers have to rely solely on the pairwise dot-products to capture interaction between the input tokens.

Given the parameter sharing and limited interactions between tokens, it is natural to wonder: what class of sequence-to-sequence functions can the Transformer networks represent? Also, what is the role of the two different kinds of layers? Are both layers needed to obtain the representation power of Transformers? In the existing literature, the advantage of Transformers has often been attributed to their capability of computing *contextual* embeddings/mappings of the input, as opposed to fixed word embeddings as in word2vec (Mikolov et al., 2013). Is it possible to formalize the notion of

---
[*]Based on work performed at Google Research New York

contextual mappings? If yes, can Transformers actually compute such mappings? Such questions still remain elusive.

In this paper, we provide a mathematical definition of contextual mappings and show that multi-head self-attention layers can indeed compute contextual mappings of the input sequences. We further show that this ability to compute contextual mappings coupled with the value mapping ability of the feed-forward layers makes Transformers universal approximators of any permutation equivariant sequence-to-sequence function. We also improve this result using positional encodings, and show that Transformers can represent any sequence-to-sequence function; i.e., the restriction of permutation equivariance can be removed by positional encodings.

These results on universal approximation of sequence-to-sequence functions raise a natural question: is it possible to have a more efficient architecture to compute contextual mappings, consequently, preserving the ability to universally approximate sequence-to-sequence functions? Towards this, we explore other architectures that can implement contextual mappings (to some extent), and experimentally evaluate their performance. In our experiments, we notice that the models that combine these simpler architectures with Transformers have better performance, compared to the standalone Transformers. We conclude the paper by presenting more discussion and interesting future research directions along these lines.

## 1.1 SUMMARY OF OUR CONTRIBUTIONS

- We prove that Transformers are universal approximators of continuous and permutation equivariant sequence-to-sequence functions with compact support (Theorem 2). We also show that, if Transformers have trainable positional encodings added to the input, then they are universal approximators of continuous sequence-to-sequence functions on a compact domain (Theorem 3).

- We formalize the notion of *contextual mappings* and show that the attention layers can compute contextual mappings, where each unique context is mapped to a unique vector (Lemma 6).

- We experimentally evaluate other simpler layers that can compute contextual mappings to some extent, such as bi-linear projections and separable convolutions, and show that substituting some of the self-attention layers with these layers can result in better performance (Section 5).

## 1.2 RELATED WORKS & NOTATION

**Analysis of attention-based models.** Given the popularity of Transformers, there have been numerous works trying to understand the role of attention layers in natural language processing models. One such line of work focuses on probing the output of attention layers to understand the attention mechanism and internal language representation (Hewitt & Manning, 2019; Clark et al., 2019; Coenen et al., 2019; Vig & Belinkov, 2019). Although these results give valuable insights, a consistent theoretical analysis corroborating these findings is missing.

**Universal approximation theorems.** Universal approximation theorems are classical results in neural network theory, dating back many decades (Cybenko, 1989; Hornik, 1991). These results show that given unbounded width, a one-hidden-layer neural network can approximate arbitrary continuous function with compact support, up to any accuracy. Other results focusing on depth appeared more recently (Lu et al., 2017; Hanin & Sellke, 2017; Lin & Jegelka, 2018). In particular, Lu et al. (2017); Hanin & Sellke (2017) consider fully-connected ReLU networks whose input dimension is $d$, and show that networks with width $d + 1$ and unbounded depth are universal approximators of scalar-valued continuous functions. Lin & Jegelka (2018) show that a residual network with one hidden neuron per residual block is a universal approximator of scalar-valued functions, given unbounded depth. Although Transformer networks do have residual connections, due to their heavy parameter sharing, the existing analyses for residual networks do not extend to Transformers. Sannai et al. (2019) consider universally approximating permutation invariant/equivariant functions using fully-connected ReLU networks.

**Turing completeness results on Transformers.** Recently, Pérez et al. (2019) have shown that Transformers with infinite precision are Turing complete, which is not the case in finite precision setting (Dehghani et al., 2018). We note that Turing completeness deals with computation on formal languages (thus discrete objects), while universal approximation focuses on functions on a continuum. In other words, these are two different concepts; and one does not imply another.

**Notation.** We use the following notation in the paper. Given a matrix $\boldsymbol{A}$, let $A_{i,j}$, $\boldsymbol{A}_{i,:}$, and $\boldsymbol{A}_{:,j}$ denote its $(i, j)$-th entry, $i$-th row, and $j$-th column, respectively. We use $\|\boldsymbol{A}\|_p$ to denote the entry-wise $\ell^p$ norm of $\boldsymbol{A}$. Let $\sigma[\cdot]$ be the softmax operator, which takes a matrix as input and applies softmax operation to each column of the matrix, which results in a column stochastic matrix, i.e., a matrix that has non-negative entries with each column summing to 1. We similarly define $\sigma_{\mathrm{H}}[\cdot]$ to be the hardmax operator, which outputs the one-hot representation of the $\arg\max$ entry for each column of the input matrix. If there are $k$ $\arg\max$ entries, then the output is $1/k$ for such entries. We use $\mathbf{1}_n$ to denote a vector of length $n$ whose entries are all 1. We denote the 0-1 indicator function by $\mathbb{1}\{\cdot\}$. We use $d$ and $n$ to denote the embedding dimension and the sequence length, respectively. We assume throughout that $n \geq 2$, as the Transformers reduce to residual networks when $n = 1$.

## 2 TRANSFORMER NETWORKS

A Transformer block is a sequence-to-sequence function mapping $\mathbb{R}^{d\times n}$ to $\mathbb{R}^{d\times n}$. It consists of two layers: a self-attention layer and a token-wise feed-forward layer, with both layers having a skip connection. More concretely, for an input $\boldsymbol{X} \in \mathbb{R}^{d\times n}$ consisting of $d$-dimensional embeddings of $n$ tokens, a Transformer block with *multiplicative* or *dot-product* attention (Luong et al., 2015) consists of the following two layers[1]:

$$\mathrm{Attn}(\boldsymbol{X}) = \boldsymbol{X} + \sum_{i=1}^{h} \boldsymbol{W}_O^i \boldsymbol{W}_V^i \boldsymbol{X} \cdot \sigma[(\boldsymbol{W}_K^i \boldsymbol{X})^T \boldsymbol{W}_Q^i \boldsymbol{X}], \tag{1}$$

$$\mathrm{FF}(\boldsymbol{X}) = \mathrm{Attn}(\boldsymbol{X}) + \boldsymbol{W}_2 \cdot \mathrm{ReLU}(\boldsymbol{W}_1 \cdot \mathrm{Attn}(\boldsymbol{X}) + \boldsymbol{b}_1 \mathbf{1}_n^T) + \boldsymbol{b}_2 \mathbf{1}_n^T, \tag{2}$$

where $\boldsymbol{W}_O^i \in \mathbb{R}^{d\times m}$, $\boldsymbol{W}_V^i, \boldsymbol{W}_K^i, \boldsymbol{W}_Q^i \in \mathbb{R}^{m\times d}$, $\boldsymbol{W}_2 \in \mathbb{R}^{d\times r}$, $\boldsymbol{W}_1 \in \mathbb{R}^{r\times d}$, $\boldsymbol{b}_2 \in \mathbb{R}^d$, $\boldsymbol{b}_1 \in \mathbb{R}^r$, and $\mathrm{FF}(\boldsymbol{X})$ is the output of the Transformer block. The number of heads $h$ and the head size $m$ are two main parameters of the attention layer; and $r$ denotes the hidden layer size of the feed-forward layer.

Here, we would like to point out that our definition of the self-attention layer (1) is an equivalent reformulation of (Vaswani et al., 2017), where they concatenate attention heads and multiply a matrix $\boldsymbol{W}_O \in \mathbb{R}^{d\times mh}$ to the concatenation. One difference in our setup is the absence of layer normalization, which simplies our analysis while preserving the basic architecture of the Transformer.

We define the Transformer networks as the composition of Transformer blocks. The family of the sequence-to-sequence functions corresponding to the Transformers can be defined as:

$$\mathcal{T}^{h,m,r} := \{g : \mathbb{R}^{d\times n} \to \mathbb{R}^{d\times n} \mid g \text{ is a composition of Transformer blocks } t^{h,m,r}\text{'s}\}.$$

where $t^{h,m,r} : \mathbb{R}^{d\times n} \to \mathbb{R}^{d\times n}$ denotes a Transformer block defined by an attention layer with $h$ heads of size $m$ each, and a feed-forward layer with $r$ hidden nodes.

We say that a function $f : \mathbb{R}^{d\times n} \to \mathbb{R}^{d\times n}$ is *permutation equivariant* if for any permutation matrix $\boldsymbol{P}$, we have $f(\boldsymbol{X}\boldsymbol{P}) = f(\boldsymbol{X})\boldsymbol{P}$; i.e., if we permute the columns of $\boldsymbol{X}$, then the columns of $f(\boldsymbol{X})$ are permuted in the same way. A Transformer block is permutation equivariant, which we formally prove in Section A. This consequently establishes the permutation equivariance of the class $\mathcal{T}^{h,m,r}$.

**Claim 1.** *A Transformer block $t^{h,m,r}$ defines a permutation equivariant map from $\mathbb{R}^{d\times n}$ to $\mathbb{R}^{d\times n}$.*

As seen in above, both layers (cf. (1) and (2)) of a Transformer block employ parameter reuse/sharing, because each token/column undergoes the same transformations (e.g., $\boldsymbol{W}_Q^i$, $\boldsymbol{W}_K^i$, or $\boldsymbol{W}_1$) regardless of its position. Moreover, interactions between tokens can only be captured through pairwise dot-products in the softmax operator $\sigma[\cdot]$ (cf. (1)). Given such limitations in a single Transformer block's representation power, it is not obvious what kinds of sequence-to-sequence functions $\mathcal{T}^{h,m,r}$ can approximate; we provide the answer to this question in the next section.

## 3 TRANSFORMERS ARE UNIVERSAL APPROXIMATORS OF SEQUENCE-TO-SEQUENCE FUNCTIONS

In this section, we present our theorems showing that the Transformer networks are universal approximators of sequence-to-sequence functions. Let us start by defining the target function class $\mathcal{F}_{\mathrm{PE}}$, which consists of all continuous permutation equivariant functions with compact support that

---

[1]In our proof we use bias vectors $\boldsymbol{b}_Q^i$ for query projections in attention layers. We omit them here for brevity.

map $\mathbb{R}^{d \times n}$ to $\mathbb{R}^{d \times n}$. Here, continuity is defined with respect to any entry-wise $\ell^p$ norm, $1 \leq p < \infty$. Given two functions $f_1, f_2 : \mathbb{R}^{d \times n} \to \mathbb{R}^{d \times n}$, for $1 \leq p < \infty$, we define a distance between them as

$$\mathsf{d}_p(f_1, f_2) := \left( \int \|f_1(\boldsymbol{X}) - f_2(\boldsymbol{X})\|_p^p \, d\boldsymbol{X} \right)^{1/p}.$$

The following result shows that a Transformer network with a constant number of heads $h$, head size $m$, and hidden layer of size $r$ can approximate any function in $\mathcal{F}_{\mathrm{PE}}$.

**Theorem 2.** *Let $1 \leq p < \infty$ and $\epsilon > 0$, then for any given $f \in \mathcal{F}_{\mathrm{PE}}$, there exists a Transformer network $g \in \mathcal{T}^{2,1,4}$, such that $\mathsf{d}_p(f, g) \leq \epsilon$.*

Next, we present our theorem on Transformers with *positional encodings*. In order to endow the Transformer networks with the ability to capture the information about the position of tokens in the input sequence, it is a common practice to add positional encodings $\boldsymbol{E} \in \mathbb{R}^{d \times n}$ to the input sequence before feeding it to the Transformer network (Vaswani et al., 2017; Devlin et al., 2018). Consider the functions represented by Transformers with positional encodings:

$$\mathcal{T}_{\mathrm{P}}^{h,m,r} := \{g_{\mathrm{P}}(\boldsymbol{X}) = g(\boldsymbol{X} + \boldsymbol{E}) \mid g \in \mathcal{T}^{h,m,r} \text{ and } \boldsymbol{E} \in \mathbb{R}^{d \times n}\}.$$

Here we show that if $\boldsymbol{E}$ is trainable, these positional encodings are sufficient to remove the permutation equivariance restriction of the Transformers. Towards this, we define $\mathcal{F}_{\mathrm{CD}}$ to be the set of all continuous functions that map a compact domain in $\mathbb{R}^{d \times n}$ to $\mathbb{R}^{d \times n}$. Note that $\mathcal{F}_{\mathrm{CD}}$ does not have the restriction of permutation equivariance as in $\mathcal{F}_{\mathrm{PE}}$, but any $f \in \mathcal{F}_{\mathrm{CD}}$ is defined on a compact domain instead of the whole $\mathbb{R}^{d \times n}$. The following result states that, equipped with the trainable positional encodings, Transformers can approximate any sequence-to-sequence function in $\mathcal{F}_{\mathrm{CD}}$.

**Theorem 3.** *Let $1 \leq p < \infty$ and $\epsilon > 0$, then for any given $f \in \mathcal{F}_{\mathrm{CD}}$, there exists a Transformer network $g \in \mathcal{T}_{\mathrm{P}}^{2,1,4}$ such that we have $\mathsf{d}_p(f, g) \leq \epsilon$.*

Theorems 2 and 3 provide an interesting characterization of the representation power of fixed-width Transformer networks. Since the function classes $\mathcal{T}^{h,m,r}$ and $\mathcal{T}_{\mathrm{P}}^{h,m,r}$ become richer as we increase the values of $(h, m, r)$, our results establish that general Transformer networks are also universal approximators of sequence-to-sequence functions. Remarkably, none of the parameters $(h, m, r)$ depend on the input sequence length $n$ or embedding dimension $d$.

Here, we would like to again point out that Theorems 2 and 3 appear quite surprising at a first glance, given the parameter sharing across all the tokens in a sequence, e.g., feed-forward layers are applied token-wise and the projection matrices in the self-attention layers are the same across different tokens. Furthermore, attention layers can only capture pairwise interaction between different tokens in the sequence. In the next subsection, we briefly describe one of our key steps in overcoming the aforementioned restrictions and proving universal approximation power of Transformers.

### 3.1 A KEY STEP: SELF-ATTENTION LAYERS CAN IMPLEMENT CONTEXTUAL MAPPINGS

Let us consider a setting where we are interested in embedding two sentences: 1) I am happy; and 2) I am Bob. These sentences are fed to a sequence-to-sequence model as

$$\boldsymbol{X} = [\boldsymbol{X}_{:,1}, \boldsymbol{X}_{:,2}, \boldsymbol{X}_{:,3}] = [\boldsymbol{v}_{\mathrm{I}}, \boldsymbol{v}_{\mathrm{am}}, \boldsymbol{v}_{\mathrm{happy}}] \text{ and } \tilde{\boldsymbol{X}} = [\tilde{\boldsymbol{X}}_{:,1}, \tilde{\boldsymbol{X}}_{:,2}, \tilde{\boldsymbol{X}}_{:,3}] = [\boldsymbol{v}_{\mathrm{I}}, \boldsymbol{v}_{\mathrm{am}}, \boldsymbol{v}_{\mathrm{Bob}}],$$

where $\boldsymbol{v}_{\mathrm{I}}, \boldsymbol{v}_{\mathrm{am}}, \boldsymbol{v}_{\mathrm{happy}}$, and $\boldsymbol{v}_{\mathrm{Bob}}$ denote $d$-dimensional embedding for the tokens 'I', 'am', 'happy', and 'Bob', respectively. Since the word 'I' occurs in different contexts in these sentences, in order to implement arbitrary sequence-to-sequence functions, the sequence-to-sequence model should map the two occurrences of 'I' to different values. We formally define this requirement below.

**Definition 3.1** (Contextual mapping). *Consider a finite set $\mathbb{L} \subset \mathbb{R}^{d \times n}$. A map $q : \mathbb{L} \to \mathbb{R}^{1 \times n}$ defines a* contextual mapping *if the map satisfies the following:*

1. *For any $\boldsymbol{L} \in \mathbb{L}$, the $n$ entries in $q(\boldsymbol{L})$ are all distinct.*
2. *For any $\boldsymbol{L}, \boldsymbol{L}' \in \mathbb{L}$, with $\boldsymbol{L} \neq \boldsymbol{L}'$, all entries of $q(\boldsymbol{L})$ and $q(\boldsymbol{L}')$ are distinct.*

In other words, a contextual mapping maps each token (column) of $\boldsymbol{L} \in \mathbb{L}$ to a unique value which depends on the entire $\boldsymbol{L}$; as a result, capturing the precise context of $\boldsymbol{L}$. This allows the subsequent

token-wise function (e.g., defined by the feed-forward layers in case of Transformer networks) to realize the outputs of any arbitrary sequence-to-sequence functions.

At the first thought, we can consider getting a contextual mapping by simply averaging all the tokens, because this can capture the one-word difference (e.g., "happy" vs. "Bob") in two different contexts. However, if there are multiple words that are different, it is not guaranteed that the average will be different. Indeed, requiring unique mappings for all the tokens for any change in any number of tokens, is a steep requirement.

While the self-attention layer does consider *pair-wise* interactions among different input tokens, it is not clear if this weak form of pair-wise interaction with shared projection weights is sufficient to extract the underlying context. The following result, which we sketch here, shows that self-attention layers can implement a *permutation equivariant* contextual mapping over almost all elements of a grid in $[0, 1]^{d \times n}$. We defer the full statement to Section 4.2.

**Lemma 6** (informal). *Consider the grid* $\mathbb{G}_\delta := \{0, \delta, \dots, 1 - \delta\}^{d \times n}$*. Then, there exist a function* $g_c : \mathbb{R}^{d \times n} \to \mathbb{R}^{d \times n}$ *composed of* $\delta^{-d} + 1$ *self-attention layers* $(h = 2, m = 1)$ *and a vector* $\boldsymbol{u} \in \mathbb{R}^d$ *such that* $q(\boldsymbol{L}) := \boldsymbol{u}^T g_c(\boldsymbol{L})$ *satisfies the following properties, for a subset* $\widetilde{\mathbb{G}}_\delta \subset \mathbb{G}_\delta$ *that contains almost all elements of* $\mathbb{G}_\delta$*:*

1. *For any* $\boldsymbol{L} \in \widetilde{\mathbb{G}}_\delta$*, the entries of* $q(\boldsymbol{L})$ *are all distinct.*

2. *For any* $\boldsymbol{L}, \boldsymbol{L}' \in \widetilde{\mathbb{G}}_\delta$ *such that* $\boldsymbol{L}$ *is not a permutation of* $\boldsymbol{L}'$*, all entries of* $q(\boldsymbol{L})$*,* $q(\boldsymbol{L}')$ *are distinct.*

Lemma 6 shows that a series of self-attention layers can implement contextual mappings, despite the apparent restriction that each of them can only capture pair-wise interaction. However, the restriction of permutation equivariance still exists because attention layers are inherently permutation equivariant. Coupled with the ability of token-wise feed-forward layers to map different values in $q(\boldsymbol{L})$ to arbitrary output values, we can prove universal approximation capability of Transformers.

### 3.2 Proof of the universal approximation theorem (Theorem 2)

Next, we outline the proof of Theorem 2 in greater detail. We refer the reader to Section C for the proof of Theorem 3, since it is a modification of Theorem 2. Even though Theorems 2 and 3 do not specifically mention the required depth for approximation, our proof techniques do characterize it, and we show that our construction is tight in the number of parameters. We defer the discussion of depth to Section 4.4.

Recall that we want to show that given a function $f \in \mathcal{F}_{\text{PE}}$, we can find a Transformer network $g \in \mathcal{T}^{2,1,4}$ such that $\mathsf{d}_p(f, g) \leq \epsilon$. Without loss of generality, we can assume that the compact support of $f$ is contained in $[0, 1]^{d \times n}$. We achieve our desired objective in three key steps:

**Step 1. Approximate $\mathcal{F}_{\text{PE}}$ with piece-wise constant functions.** We first use (a variant of) the classical result that any continuous function can be approximated up to arbitrary accuracy by piece-wise constant functions. For $\delta > 0$, we define the following class of piece-wise constant functions.

$$\overline{\mathcal{F}}_{\text{PE}}(\delta) := \left\{ f : \boldsymbol{X} \mapsto \sum\nolimits_{\boldsymbol{L} \in \mathbb{G}_\delta} \boldsymbol{A}_{\boldsymbol{L}} \mathbb{1}\left\{ \boldsymbol{X} \in \mathbb{S}_{\boldsymbol{L}} \right\} \mid f \text{ is permutation equivariant, } \boldsymbol{A}_{\boldsymbol{L}} \in \mathbb{R}^{d \times n} \right\},$$

where $\mathbb{G}_\delta := \{0, \delta, \dots, 1 - \delta\}^{d \times n}$ and, for a grid point $\boldsymbol{L} \in \mathbb{G}_\delta$, $\mathbb{S}_{\boldsymbol{L}} := \prod_{j=1}^d \prod_{k=1}^n [L_{j,k}, L_{j,k} + \delta) \subset [0, 1]^{d \times n}$ denotes the associated cube of width $\delta$. Let $\overline{f} \in \overline{\mathcal{F}}_{\text{PE}}(\delta)$ be such that $\mathsf{d}_p(f, \overline{f}) \leq \epsilon/3$.

**Step 2. Approximate $\overline{\mathcal{F}}_{\text{PE}}(\delta)$ with *modified* Transformers.** We then consider a slightly modified architecture for Transformer networks, where the softmax operator $\sigma[\cdot]$ and $\text{ReLU}(\cdot)$ are replaced by the hardmax operator $\sigma_{\text{H}}[\cdot]$ and an activation function $\phi \in \Phi$, respectively. Here, the set of allowed activations $\Phi$ consists of all piece-wise linear functions with at most three pieces, where at least one piece is constant. Let $\overline{\mathcal{T}}^{h,m,r}$ denote the function class corresponding to the sequence-to-sequence functions defined by the modified Transformer networks. The following result establishes that the modified Transformer networks in $\overline{\mathcal{T}}^{2,1,1}$ can closely approximate functions in $\overline{\mathcal{F}}_{\text{PE}}(\delta)$.

**Proposition 4.** *For each* $\overline{f} \in \overline{\mathcal{F}}_{\text{PE}}(\delta)$ *and* $1 \leq p < \infty$*,* $\exists\, \overline{g} \in \overline{\mathcal{T}}^{2,1,1}$ *such that* $\mathsf{d}_p(\overline{f}, \overline{g}) = O(\delta^{d/p})$*.*

**Step 3. Approximate modified Transformers with (original) Transformers.** Finally, we show that $\overline{g} \in \overline{\mathcal{T}}^{2,1,1}$ can be approximated by $\mathcal{T}^{2,1,4}$. Let $g \in \mathcal{T}^{2,1,4}$ be such that $\mathsf{d}_p(\overline{g}, g) \leq \epsilon/3$.

Theorem 2 now follows from these three steps, because we have

$$\mathsf{d}_p(f, g) \leq \mathsf{d}_p(f, \overline{f}) + \mathsf{d}_p(\overline{f}, \overline{g}) + \mathsf{d}_p(\overline{g}, g) \leq 2\epsilon/3 + O(\delta^{d/p}).$$

Choosing $\delta$ small enough ensures that $\mathsf{d}_p(f, g) \leq \epsilon$. □

We refer the reader to Sections B.1 and B.2 in the supplementary material for the formal statements and proofs of Steps 1 and 3, respectively. As for Step 2, which is the most critical step in establishing the universal approximation property of Transformers, we provide a sketch of the proof of Proposition 4 in the next section, and refer the reader to Section B.3 for the complete proof.

## 4 PROOF SKETCH OF PROPOSITION 4: DIFFERENT ROLES OF TWO LAYERS

As mentioned earlier, the heavy parameter sharing in Transformers makes the goal of universally approximating sequence-to-sequence functions seemingly difficult. Both the self-attention and the feed-forward layer weights inside a Transformer block are fixed across $n$ tokens. In this section, we show that Transformers are able to overcome this architectural constraint, and compute contextual mappings of the entire input sequence just based on the pair-wise interactions. The token-wise feedforward layers then transform these contextual mappings to the desired output sequence.

We highlight these inner workings of Transformers en route to proving Proposition 4. We want to show that given a piece-wise constant function $\overline{f} \in \overline{\mathcal{F}}_{PE}(\delta)$, there exists a modified Transformer network $\overline{g} \in \overline{\mathcal{T}}^{2,1,1}$ that closely approximates $\overline{f}$. We achieve this goal by establishing the following three claims, which correspond to Lemmas 5, 6, and 7.

1. Given an input $\boldsymbol{X} \in \mathbb{R}^{d \times n}$, a series of feed-forward layers in the modified Transformer network can quantize $\boldsymbol{X}$ to an element $\boldsymbol{L}$ on the extended grid $\mathbb{G}_\delta^+ := \{-\delta^{-nd}, 0, \delta, \dots, 1 - \delta\}^{d \times n}$.

2. Next, a series of self-attention layers in the modified Transformer network can take the input $\boldsymbol{L}$ and implement a *contextual mapping* $q$ such that, for $\boldsymbol{L}$ and $\boldsymbol{L}'$ that are not permutation of each other, all the elements in $q(\boldsymbol{L})$ and $q(\boldsymbol{L}')$ are distinct.

3. Finally, a series of feed-forward layers in the modified Transformer network can map elements of the contextual embedding $q(\boldsymbol{L})$ to the desired output value of $\overline{f} \in \overline{\mathcal{F}}_{\mathrm{PE}}$ at the input $\boldsymbol{X}$.

Before discussing these three claims in detail, we note that even though a Transformer network stacks self-attention and feed-forward layers in an alternate manner, the skip connections enable these networks to employ a composition of multiple self-attention or feed-forward layers. Furthermore, as alluded earlier, these three steps clearly highlight the different roles that self-attention and feed-forward layers play in realizing the ability to universally approximate sequence-to-sequence functions: 1) self-attention layers compute precise contextual maps; and 2) feed-forward layers then assign the results of these contextual maps to the desired output values.

### 4.1 QUANTIZATION BY FEED-FORWARD LAYERS

Since our objective in Proposition 4 is to approximate the function $\overline{f} \in \overline{\mathcal{F}}_{\mathrm{PE}}(\delta)$, which takes a constant value on the cubes $\mathbb{S}_{\boldsymbol{L}}$'s, the (modified) Transformer network approximating $\overline{f}$ first quantizes the input $\boldsymbol{X}$ according to these cubes. In particular, we want each input $\boldsymbol{X} \in \mathbb{S}_{\boldsymbol{L}}$ to be mapped to the point $\boldsymbol{L}$. The following result shows that a modified Transformer network can indeed implement this quantization map with a composition of multiple feed-forward layers.

**Lemma 5.** *Consider a scalar quantization map $g_q^{\mathrm{ent}} : \mathbb{R} \to \{-\delta^{-nd}, 0, \delta, \dots, 1 - \delta\}$:*

$$g_{\mathrm{q}}^{\mathrm{ent}}(t) = \begin{cases} k\delta & \text{if } k\delta \leq t < (k+1)\delta, \ k = 0, \dots, 1/\delta - 1, \\ -\delta^{-nd} & \text{otherwise.} \end{cases}$$

*There exists a function $g_{\mathrm{q}} : \mathbb{R}^{d \times n} \mapsto \mathbb{G}_\delta^+$ composed of $\frac{d}{\delta} + d$ token-wise feed-forward layers with $r = 1$ and activations in $\Phi$, which employs the scalar quantization $g_q^{\mathrm{ent}}$ to each entry of its input.*

As desired, the function $g_{\mathrm{q}}$ maps any $\boldsymbol{X} \in \mathbb{S}_{\boldsymbol{L}}$ to $\boldsymbol{L}$. Furthermore, if any element of $\boldsymbol{X}$ is not in $[0, 1]$, the element is mapped to $-\delta^{-nd}$, indicating that $\boldsymbol{X}$ is outside the compact support of $\overline{f} \in \overline{\mathcal{F}}_{\mathrm{PE}}(\delta)$.

## 4.2 CONTEXTUAL MAPPING BY SELF-ATTENTION LAYERS

In this subsection, we show that the (modified) Transformer network can compute contextual mappings (cf. Definition 3.1) from the output $L \in \mathbb{G}_\delta^+$ of the map $g_q$ (cf. Section 4.1) by using a composition of self-attention layers. The following lemma, sketched earlier in Section 3.1, shows that the (modified) Transformer networks can implement a *permutation equivariant* contextual mapping over almost all elements of $\mathbb{G}_\delta$, while mapping the rest of elements in $\mathbb{G}_\delta^+$ to a disjoint set.

**Lemma 6.** *Consider the following subset of* $\mathbb{G}_\delta = \{0, \delta, \dots, 1 - \delta\}^{d \times n}$:

$$\widetilde{\mathbb{G}}_\delta := \{L \in \mathbb{G}_\delta \mid L_{:,i} \neq L_{:,j} \text{ for all } i \neq j\}.$$

*Assume that* $n \geq 2$ *and* $\delta^{-1} \geq 2$. *Then, there exist a function* $g_c : \mathbb{R}^{d \times n} \to \mathbb{R}^{d \times n}$ *composed of* $\delta^{-d} + 1$ *self-attention layers* ($h = 2, m = 1$) *that employ the* $\sigma_H$ *operator, a vector* $u \in \mathbb{R}^d$, *constants* $t_l, t_r \in \mathbb{R}$ ($0 < t_l < t_r$)*, such that* $q(L) := u^T g_c(L)$ *satisfies the following properties:*

1. *For any* $L \in \widetilde{\mathbb{G}}_\delta$, *the entries of* $q(L)$ *are all distinct.*

2. *For any* $L, L' \in \widetilde{\mathbb{G}}_\delta$ *such that* $L$ *is not a permutation of* $L'$, *all entries of* $q(L)$, $q(L')$ *are distinct.*

3. *For any* $L \in \widetilde{\mathbb{G}}_\delta$, *all the entries of* $q(L)$ *are in* $[t_l, t_r]$.

4. *For any* $L \in \mathbb{G}_\delta^+ \setminus \widetilde{\mathbb{G}}_\delta$, *all the entries of* $q(L)$ *are outside* $[t_l, t_r]$.

At this point, a few remarks about the result in Lemma 6 are in order. First, since the Transformer networks are bound to implement permutation invariant maps, we require the Property 6.2 to hold for the pair of sequences that cannot be mapped to each other via permutation of columns. Furthermore, the self-attention layers implement the desirable contextual map for only $\widetilde{\mathbb{G}}_\delta \subseteq \mathbb{G}_\delta$, where all columns of $L$ are distinct. Note that for small $\delta$, $\mathbb{G}_\delta \setminus \widetilde{\mathbb{G}}_\delta$ constitutes a negligible fraction of $\mathbb{G}_\delta$ because $|\mathbb{G}_\delta \setminus \widetilde{\mathbb{G}}_\delta| = O(\delta^d |\mathbb{G}_\delta|)$. The function $q$ in Lemma 6 maps the elements of $\mathbb{G}_\delta^+ \setminus \widetilde{\mathbb{G}}_\delta$ outside $[t_l, t_r]$—the interval where the outputs of the contextual mapping for $\widetilde{\mathbb{G}}_\delta$ reside.

### 4.2.1 PROOF SKETCH OF LEMMA 6

Since Lemma 6 is one of the major technical contributions of this paper, we provide a short sketch of its proof. The complete proof is presented in Section B.5. For simplicity, we consider the case $d = 1$, so the input $L \in \mathbb{G}_\delta^+$ is a row vector of length $n$.

The key idea of the proof is that, using two attention heads of size 1, one can implement a self-attention layer that shifts up input entries that are *in a specific interval*, while leaving all other entries intact. We call this the **selective shift operation**. Since the entries in $L$ are quantized, we apply the selective shift operation to $0, \delta, \dots, 1 - \delta$ using $1/\delta$ attention layers. Interestingly, the value of the largest output entry after these operations is unique for each $L \in \widetilde{\mathbb{G}}_\delta$ up to permutations. Using the largest entry, one can add one last layer that shifts up the entire matrix and outputs $q(L)$ that satisfies Properties 6.1 and 6.2 of the lemma.

More concretely, the following function $\Psi : \mathbb{R}^{1 \times n} \to \mathbb{R}^{1 \times n}$, parametrized by $b, b' \in \mathbb{R}$ satisfying $b < b'$, can be implemented with two attention heads of size 1 with the hardmax ($\sigma_H$) operator:

$$\Psi(Z; b, b')_{1,j} = \begin{cases} \max_k Z_{1,k} - \min_k Z_{1,k} & \text{if } b < Z_{1,j} < b', \\ 0 & \text{if } Z_{1,j} < b \text{ or } Z_{1,j} > b'. \end{cases}$$

If we define an attention layer of the form $Z \mapsto Z + \Psi(Z; b, b')$, then any entry $Z_{1,j}$ in $(b, b')$ is shifted up by $\max_k Z_{1,k} - \min_k Z_{1,k}$, *while all the other entries stay untouched*. We can choose $b$ and $b'$ to selectively shift certain entries, hence the name selective shift operation.

We stack $1/\delta$ self-attention layers, with attention parts $\delta^{-1} \Psi(\cdot; l - \delta/2, l + \delta/2)$ for each $l \in \{0, \delta, \dots, 1 - \delta\}$, in increasing order of $l$. With these layers, we can apply the selective shift operations to input entries of values $0, \delta, \dots, 1 - \delta$. To see how the shift operations modify the input, now consider $n = 2$ for simplicity, and let $L = [l_1 \quad l_2] \in \widetilde{\mathbb{G}}_\delta$. Without loss of generality, we can assume $l_1 < l_2$. The selective shift operation is applied to $l_1$ first, shifting it by $\delta^{-1}(\max L - \min L) = \delta^{-1}(l_2 - l_1)$, resulting in $\widetilde{l}_1 = l_1 + \delta^{-1}(l_2 - l_1) > l_2$. After that, the operation on $l_2$ shifts it up by $\delta^{-1}(\widetilde{l}_1 - l_2)$. Thus, the first $1/\delta$ layers map $L = [l_1 \quad l_2]$ ($l_1 < l_2$) to

$$\widetilde{L} = [\widetilde{l}_1 \quad \widetilde{l}_2] := [l_1 + \delta^{-1}(l_2 - l_1) \quad l_2 + (\delta^{-2} - \delta^{-1})(l_2 - l_1)].$$

We can show that the map from $[l_1 \quad l_2] \in \{ \boldsymbol{L} \in \widetilde{\mathbb{G}}_\delta \mid l_1 < l_2 \}$ to $\widetilde{l}_2$ is one-to-one, and that $0 < \widetilde{l}_1 < \widetilde{l}_2 < \delta^{-2}$. We then add one last layer that shifts all positive entries of $\widetilde{\boldsymbol{L}}$ by $\delta^{-3} \max \widetilde{\boldsymbol{L}} = \delta^{-3} \widetilde{l}_2$, whose output we denote by $q(\boldsymbol{L}) = [\delta^{-3} \widetilde{l}_2 + \widetilde{l}_1 \quad \delta^{-3} \widetilde{l}_2 + \widetilde{l}_2]$. All entries of $q(\boldsymbol{L})$ are in $[\delta^{-3} \widetilde{l}_2, \delta^{-3} \widetilde{l}_2 + \delta^{-2})$, and this interval is disjoint for different $\boldsymbol{L}$'s because $\boldsymbol{L} \mapsto \widetilde{l}_2$ is one-to-one. Thus, $q(\boldsymbol{L})$ satisfies Properties 6.1 and 6.2 of the lemma. The remaining details are in Section B.5.

### 4.3 FUNCTION VALUE MAPPING BY FEED-FORWARD LAYERS

This brings us to the final step, which demonstrates the key utility of the feed-forward layers. After the contextual mapping by self-attention layers, each token captures the entire context available in the input sequence. The following result shows that token-wise application of a composition of feed-forward layers can map these tokens to the desired output values required by the function $\overline{f}$.

**Lemma 7.** *Let* $g_c : \mathbb{R}^{d \times n} \to \mathbb{R}^{d \times n}$ *be the function from Lemma 6. Then, there exists a function* $g_v : \mathbb{R}^{d \times n} \to \mathbb{R}^{d \times n}$ *composed of* $O(n(\frac{1}{\delta})^{dn}/n!)$ *token-wise feed-forward layers* $(r = 1)$ *with activations in* $\Phi$ *such that* $g_v$ *is defined by a token-wise function* $g_v^{\text{tkn}} : \mathbb{R}^d \to \mathbb{R}^d$ *on each column,*

$$g_v(\boldsymbol{Z}) = [g_v^{\text{tkn}}(\boldsymbol{Z}_{:,1}) \quad \cdots \quad g_v^{\text{tkn}}(\boldsymbol{Z}_{:,n})],$$

*where for all* $j \in \{1, \ldots, n\}$,

$$g_v^{\text{tkn}}(g_c(\boldsymbol{L})_{:,j}) = \begin{cases} (\boldsymbol{A}_L)_{:,j} & \text{if } \boldsymbol{L} \in \widetilde{\mathbb{G}}_\delta, \\ \boldsymbol{0}_d & \text{if } \boldsymbol{L} \in \mathbb{G}_\delta^+ \setminus \widetilde{\mathbb{G}}_\delta. \end{cases}$$

### 4.4 TIGHTNESS OF CONSTRUCTIONS

We showed in this section that Theorem 2 requires $O(n(1/\delta)^{dn}/n!)$ Transformer blocks for approximation, where $\delta$ is the width of the cubes. Each transformer block is of constant width, so it has $O(d)$ parameters; this means that the total number of parameters is $O(dn(1/\delta)^{dn}/n!)$. We note that this exponential dependence cannot be avoided in the worse case. If we assume continuity without any additional smoothness, quantizing the domain to cubes and approximating the function with constants require memorizing (output dim) $\times$ (num cubes)$/n!$ real numbers, where the factor of $1/n!$ is due to permutation equivariance. Thus, Theorem 2 is optimal in the order of parameters.

If we compare with the residual network result (Lin & Jegelka, 2018), we can consider "flattening" $\boldsymbol{X}$ into a $dn$-dimensional vector and fitting the function. The proof technique in (Lin & Jegelka, 2018) requires $O((1/\delta)^{dn})$ layers, where each layer has $O(dn)$ parameters: the total parameter requirement is $O(dn(1/\delta)^{dn})$. This shows that Transformers can approximate permutation equivariant functions in a more efficient way than residual networks.

In Section C, our proof of Theorem 3 shows that we require $O(n(1/\delta)^{dn})$ layers to approximate continuous (not permutation equivariant) sequence-to-sequence functions. As seen from the argument above, this construction is also optimal in the order of parameters.

## 5 DISCUSSION AND EXPERIMENTS

As detailed in Section 4, the ability of the self-attention layers to compute contextual mappings plays a crucial role in the universal approximation property. Interestingly, our analysis shows that replacing the dot-product attention in Transformers with any other component capable of computing contextual mappings should preserve this universal approximation property. This leads naturally to questions about the alternative architectures that realize certain kinds of contextual mappings at different computational and memory costs. We explore and discuss some examples of such alternatives in this section. Our preliminary empirical study demonstrates their practical utility.

### 5.1 BI-LINEAR PROJECTION

Given token embeddings $\boldsymbol{X}$ as input, the bi-linear projection layer computes the following update.

$$\text{BProj}(\boldsymbol{X}) = \boldsymbol{X} + \boldsymbol{W}_O \cdot \boldsymbol{X} \cdot \boldsymbol{W}_P.$$

The bi-linear projection layer (Gong et al., 2013) is motivated from the ability of random (Gaussian) matrices to map sparse differences to dense vectors (Ailon & Chazelle, 2009). If there are two input

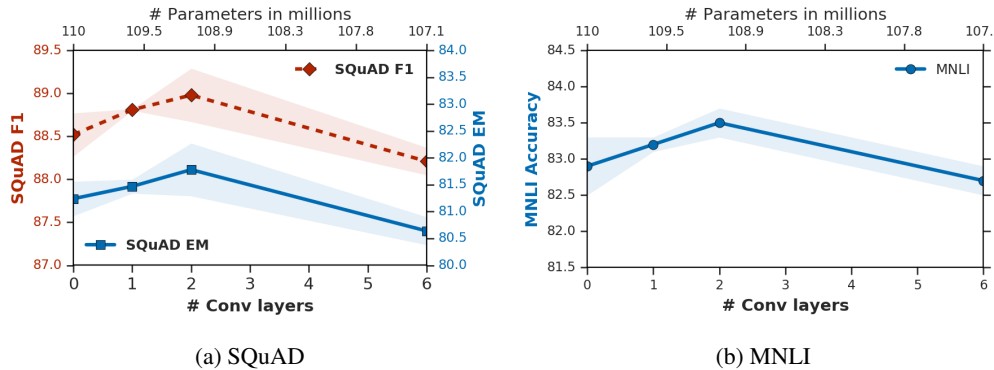

(a) SQuAD                                          (b) MNLI

Figure 1: Performance of hybrid models constructed by first taking BERT$_{\text{BASE}}$, a 12 layer Transformer model, and replacing the self-attention layers with depth-wise separable convolution layers, in a varying number of the Transformer blocks closer to the input. Surprisingly, replacing 1 or 2 self-attention layers with convolutions improves the performance, while replacing more hurts the performance. This suggests both that Transformers have functionality beyond just computing contextual mappings, and having simpler layers to realize contextual mapping can aid Transformers.

contexts $X_1$ and $X_2$ that differ in one token, their difference $X_1 - X_2$ is sparse; however, after random projection, the difference $(X_1 - X_2)W_P$ will be dense, and the numbers are distinct with high probability, implementing a form "pair-wise contextual mapping,"[2] although different from the contextual mapping in Definition 3.1.

This layer advantageously incurs smaller number of matrix multiplications as compared to the dot-product attention. That said, the number of parameters in this layer depend on the sequence length, making it harder to reuse the model across tasks with different input sequence lengths. Moreover, the weights used to compute the contextual embeddings ($W_P$) are independent of the inputs ($X$), whereas in self-attention the weights ($\sigma[(W_K^i X)^T W_Q^i X]$) depend on $X$. The first drawback can be addressed by replacing the linear projection with a depth-wise separable convolution layer, which is discussed in the next subsection.

### 5.2 DEPTH-WISE SEPARABLE CONVOLUTIONS

A depth-wise convolution layer (Sifre & Mallat, 2014; Chollet, 2017; Kaiser et al., 2017) involves convolving each dimension of $X$ with a corresponding convolution filter of size $k$:

$$\text{SepConv}(X) = X + W_O(X * W_C),$$

where $W_C \in \mathbb{R}^{d \times k}$ and $(X * W_C)_{i,:} := X_{i,:} * (W_C)_{i,:}$. Unlike bi-linear projection, this layer can be used across tasks with different input sequence lengths as the number of parameters are independent of the sequence length. While a single layer is unable to compute contextual mappings when the filter size is small, stacking multiple such layers can potentially provide a cheaper way to compute contextual mappings. In fact, based on depth-wise separable convolutions, Wu et al. (2019) proposed a light-weight dynamic convolution architecture that performs competitively with Transformers on machine translation.

### 5.3 EXPERIMENTS

We now present our experiments with these other architectures, with the goal of understanding the extent to which computing contextual mappings can capture the performance of Transformers. As discussed earlier, BProj and SepConv do *not* implement contextual mappings (cf. Definition 3.1), so we do not expect that either BProj or SepConv based models to have the same performance as the expensive Transformers. These models do not use input dependent weights to compute attention, and hence have weaker representation power. Instead, our goal is to see if we can use these cheaper layers to replace (some of) the expensive self-attention layers.

---

[2]This guarantee only holds for a finite set (can be exponential in $n$) of fixed vectors in $\mathbb{R}^n$.

We follow the experimental setting from Devlin et al. (2018) to train the Transformers, with the masked language model pre-training followed by a task specific fine-tuning, and work with a 12 layer architecture based on BERT$_{\text{BASE}}$. We present our results on a question answering task (SQuAD) (Rajpurkar et al., 2016) and a sentence entailment task (MNLI) (Williams et al., 2018). In our first set of experiments we train models that employ BProj and SepConv layers, instead of the self-attention layer in eq.(1). We notice that, as expected, these simpler models have weaker performance than the self-attention layer. See Table 1 in Section D for a comparison of these models on MNLI.

Next, we swap a varying number of the first few self-attention layers in BERT$_{\text{BASE}}$ with SepConv, implemented with filter reuse across dimensions (Wu et al., 2019)[3]. Fig. 1 illustrates the performance of these hybrid models. Interestingly, models with 1 or 2 convolution layers and rest the self-attention layers, perform better than models with only the self-attention layers. Note that, replacing self-attention layer with SepConv also reduces the computational cost and the number of parameters. One explanation we have is that the first few attention layers tend to attend broadly to the whole sequence (as empirically observed in (Clark et al., 2019)), and the cheaper convolution layers can perform this job more efficiently. A detailed evaluation of such hybrid architectures will be interesting future research.

Our experiments also call for a deeper understanding of the exact nature of the embeddings computed by practical attention models. Since Transformers in practice have fixed depth, we believe that they might not be able to exactly implement contextual mappings as we defined in Definition 3.1. However, there is some preliminary empirical evidence that Transformers do implement some sort of "contextual mappings." For example, Fig. 4 of Coenen et al. (2019) presents visualizations of embeddings of a single word in different contexts (sentences). They experimentally notice that Transformers, in addition to computing contextual mappings, also map a word into semantic clusters. Formalizing and evaluating this property of Transformers is an interesting direction for future work. We again note that Wu et al. (2019) have proposed an alternative way to compute such embeddings based on dynamic convolution layers. Evaluating the mappings computed by these models should shed more light on the workings of attention models and inspire efficient and better performing architectures.

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

# A  PROOF OF CLAIM 1

Suppose $\boldsymbol{XP}$ was given as input, where $\boldsymbol{P}$ is a permutation matrix. First note that

$$(\boldsymbol{W}_K^i \boldsymbol{XP})^T(\boldsymbol{W}_Q^i \boldsymbol{XP}) = \boldsymbol{P}^T(\boldsymbol{W}_K^i \boldsymbol{X})^T(\boldsymbol{W}_Q^i \boldsymbol{X})\boldsymbol{P}$$

After the softmax operation, we get

$$\sigma[\boldsymbol{P}^T(\boldsymbol{W}_K^i \boldsymbol{X})^T(\boldsymbol{W}_Q^i \boldsymbol{X})\boldsymbol{P}] = \boldsymbol{P}^T \sigma[(\boldsymbol{W}_K^i \boldsymbol{X})^T(\boldsymbol{W}_Q^i \boldsymbol{X})]\boldsymbol{P}.$$

Then,

$$\mathrm{Attn}(\boldsymbol{XP}) = \boldsymbol{XP} + \sum_{i=1}^h \boldsymbol{W}_O^i(\boldsymbol{W}_V^i \boldsymbol{XP}) \cdot \boldsymbol{P}^T \sigma[(\boldsymbol{W}_K^i \boldsymbol{X})^T(\boldsymbol{W}_Q^i \boldsymbol{X})]\boldsymbol{P} = \mathrm{Attn}(\boldsymbol{X})\boldsymbol{P},$$

where we used $\boldsymbol{PP}^T = \boldsymbol{I}$. Permutation equivariance of the token-wise feed-forward layer can be shown similarly:

$$\mathrm{FF}(\boldsymbol{XP}) = \mathrm{Attn}(\boldsymbol{X})\boldsymbol{P} + \boldsymbol{W}_2 \cdot \mathrm{ReLU}(\boldsymbol{W}_1 \cdot \mathrm{Attn}(\boldsymbol{X})\boldsymbol{P} + \boldsymbol{b}_1 \boldsymbol{1}_n^T \boldsymbol{P}) + \boldsymbol{b}_2 \boldsymbol{1}_n^T \boldsymbol{P}$$
$$= \mathrm{Attn}(\boldsymbol{X})\boldsymbol{P} + \boldsymbol{W}_2 \cdot \mathrm{ReLU}(\boldsymbol{W}_1 \cdot \mathrm{Attn}(\boldsymbol{X}) + \boldsymbol{b}_1 \boldsymbol{1}_n^T)\boldsymbol{P} + \boldsymbol{b}_2 \boldsymbol{1}_n^T \boldsymbol{P} = \mathrm{FF}(\boldsymbol{X})\boldsymbol{P},$$

where $\mathrm{ReLU}(\boldsymbol{XP}) = \mathrm{ReLU}(\boldsymbol{X})\boldsymbol{P}$ was used. This analysis shows that the function class $\mathcal{T}^{h,m,r}(\cdot)$ is restricted to permutation equivariant functions.

# B  PROOF DETAILS OF THEOREM 2

We first define some additional notation. For $a, b \in \mathbb{N}$ where $a \leq b$, let $[a] = \{1, \ldots, a\}$ and $[a:b] = \{a, a+1, \ldots, b-1, b\}$. For $a, b, c \in \mathbb{R}$ where $b - a > 0$ is an integer multiple of $c > 0$, we write $[a:c:b] := \{a, a+c, a+2c, \ldots, b-c, b\}$.

## B.1  APPROXIMATING $\mathcal{F}_{\mathrm{PE}}$ WITH $\overline{\mathcal{F}}_{\mathrm{PE}}(\delta)$

**Lemma 8.** *For any given $f \in \mathcal{F}_{\mathrm{PE}}$ and $1 \leq p < \infty$, one can find a $\delta^* > 0$ such that $\exists \overline{f} \in \overline{\mathcal{F}}_{\mathrm{PE}}(\delta^*)$ which satisfies $\mathsf{d}_p(f, \overline{f}) \leq \epsilon/3$.*

**Proof**  Since $f : \mathbb{R}^{d \times n} \to \mathbb{R}^{d \times n}$ is a continuous function with compact support, the function is uniformly continuous. Since continuity is defined using entry-wise $\ell_p$ norm, and entry-wise $\ell_p$ norm is equivalent to entry-wise $\ell_\infty$ norm when the number of entries are finite, uniform continuity implies that

$$\forall \epsilon > 0, \exists \delta > 0 \text{ such that } \forall \boldsymbol{X}, \boldsymbol{Y}, \|\boldsymbol{X} - \boldsymbol{Y}\|_\infty < \delta \implies \|f(\boldsymbol{X}) - f(\boldsymbol{Y})\|_p < \epsilon.$$

This means that given any $\epsilon/3 > 0$, we have such a $\delta > 0$. Using this $\delta$, we can create a grid $\mathbb{G}_\delta$ and corresponding cubes $\mathbb{S}_L$, as described in the main text. For any $\boldsymbol{L} \in \mathbb{G}_\delta$, we define $\boldsymbol{C_L} \in \mathbb{S}_L$ to be the center point of the cube $\mathbb{S}_L$. Then, we can define a piece-wise constant approximation $\overline{f}(\boldsymbol{X}) = \sum_{\boldsymbol{L} \in \mathbb{G}_\delta} f(\boldsymbol{C_L}) \mathbb{1}\{\boldsymbol{X} \in \mathbb{S}_L\}$. Note that, for any $\boldsymbol{X} \in \mathbb{S}_L$, we have $\|\boldsymbol{X} - \boldsymbol{C_L}\|_\infty < \delta$, so by uniform continuity, we have $\|f(\boldsymbol{X}) - \overline{f}(\boldsymbol{X})\|_p = \|f(\boldsymbol{X}) - f(\boldsymbol{C_L})\|_p < \epsilon/3$. This proves that $\mathsf{d}_p(f, \overline{f}) < \epsilon/3$.

As for permutation equivariance, since $f$ is permutation equivariant, we have $f(\boldsymbol{C_L P}) = f(\boldsymbol{C_L})\boldsymbol{P}$ for any permutation matrix $\boldsymbol{P}$. For any $\boldsymbol{X} \in \mathbb{S}_L$, we have $\boldsymbol{XP} \in \mathbb{S}_{LP}$, so

$$\overline{f}(\boldsymbol{XP}) = f(\boldsymbol{C_{LP}}) = f(\boldsymbol{C_L P}) = f(\boldsymbol{C_L})\boldsymbol{P} = \overline{f}(\boldsymbol{X})\boldsymbol{P}.$$

Thus, the approximation $\overline{f}$ is also permutation equivariant. This proves the lemma.  $\square$

## B.2 Approximating $\overline{\mathcal{T}}^{2,1,1}$ with $\mathcal{T}^{2,1,4}$

**Lemma 9.** *For each $\overline{g} \in \overline{\mathcal{T}}^{2,1,1}$ and $1 \leq p < \infty$, $\exists\, g \in \mathcal{T}^{2,1,4}$ such that $\mathsf{d}_p(\overline{g}, g) \leq \epsilon/3$.*

**Proof** Recall that $T^{h,m,r}$ refers to the class of functions representable with composition of Transformer blocks with $h$ heads of size $m$ in self-attention layers and $r$ hidden nodes in feed-forward layers. The same notation holds for the modified Transformers $\overline{\mathcal{T}}^{h,m,r}$.

Note that the softmax operator on a matrix $\boldsymbol{A}$ can be made arbitrarily close to hardmax by scaling up $\boldsymbol{A}$. That is,

$$\sigma[\lambda \boldsymbol{A}] \to \sigma_{\mathrm{H}}[\boldsymbol{A}] \quad \text{as } \lambda \to \infty.$$

This means that by scaling up parameters inside $\sigma$, we can approximate $\sigma_{\mathrm{H}}$ arbitrarily closely. Thus, the modified self-attention layers can be approximated with the original self-attention layers of the same number of heads $h$ and head size $m$.

Also, any arbitrary (possibly discontinuous) piecewise linear function $\phi \in \Phi$ can be approximated arbitrarily closely by four ReLU's. Note that $\phi \in \Phi$ as at most three pieces, and at least one of the pieces is constant. For example, consider the following function $\phi \in \Phi$:

$$\phi(t) = \begin{cases} b_1 & \text{if } t < c_1, \\ a_2 t + b_2 & \text{if } c_1 \leq t < c_2, \\ a_3 t + b_3 & \text{if } c_2 \leq t. \end{cases}$$

This function can be approximated by four ReLU's, as claimed by the lemma:

$$\begin{aligned}
\widetilde{\phi}(t) = \ & b_1 + \frac{a_2 c_1 + b_2 - b_1}{\epsilon} \mathrm{ReLU}(t - c_1 + \epsilon) + \left( a_2 - \frac{a_2 c_1 + b_2 - b_1}{\epsilon} \right) \mathrm{ReLU}(t - c_1) \\
& + \left( \frac{a_3 c_2 + b_3 - a_2(c_2 - \epsilon) - b_2}{\epsilon} - a_2 \right) \mathrm{ReLU}(t - c_2 + \epsilon) \\
& + \left( a_3 - \frac{a_3 c_2 + b_3 - a_2(c_2 - \epsilon) - b_2}{\epsilon} \right) \mathrm{ReLU}(t - c_2) \\
= \ & \begin{cases} b_1 & \text{if } t < c_1 - \epsilon, \\ \frac{a_2 c_1 + b_2 - b_1}{\epsilon}(t - c_1) + a_2 c_1 + b_2 & \text{if } c_1 - \epsilon \leq t < c_1, \\ a_2 t + b_2 & \text{if } c_1 \leq t < c_2 - \epsilon, \\ \frac{a_3 c_2 + b_3 - a_2(c_2 - \epsilon) - b_2}{\epsilon}(t - c_2) + a_3 c_2 + b_3 & \text{if } c_2 - \epsilon \leq t < c_2, \\ a_3 t + b_3 & \text{if } c_2 \leq t. \end{cases}
\end{aligned}$$

Also, as we make $\epsilon \to 0$, we can approximate $\phi$ as closely as possible using $\widetilde{\phi}$. The cases where the second or third piece is constant can be shown similarly. This means that the modified feed-forward layers (whose activation is $\phi \in \Phi$) with single hidden node can be approximated with the original feed-forward layers (ReLU) with four hidden nodes.

Thus, given any $\overline{g} \in \overline{\mathcal{T}}^{2,1,1}$, there exists a function $g \in \mathcal{T}^{2,1,4}$ arbitrarily close to $\overline{g}$, by appropriately choosing the parameters to be large enough. This finishes the proof. $\qquad \square$

## B.3 Finishing proof of Proposition 4

As we have already discussed in Section 4, we establish Proposition 4 in three steps:

1. Given an input $\boldsymbol{X}$, a group of feed-forward layers in the modified Transformer network can quantize $\boldsymbol{X}$ to an element $\boldsymbol{L}$ on the extended grid $\mathbb{G}_\delta^+ := \{-\delta^{-nd}, 0, \delta, \dots, 1 - \delta\}^{d \times n}$.

2. Next, a group of self-attention layers in the modified Transformer network can take the input $\boldsymbol{L}$ and produce desirable *contextual mappings* $q(\boldsymbol{L})$ such that, for $\boldsymbol{L}$ and $\tilde{\boldsymbol{L}}$, that are not permutation of each other, all the elements in $q(\boldsymbol{L})$ and $q(\tilde{\boldsymbol{L}})$ are distinct.

3. Finally, a group of feed-forward layers in the modified Transformer network can map elements of the contextual embedding $q(\boldsymbol{L})$ to the desirable values, i.e., the output of $\overline{f} \in \overline{\mathcal{F}}_{\text{PE}}$ on the input $\boldsymbol{X}$.

These steps are formally stated in Lemmas 5, 6, and 7 in the main text. We present the proofs of these lemmas in the subsequent sections.

With the results established in these lemmas, we are now equipped with all the tools necessary to complete the proof of Proposition 4. Let us recall the functions $g_{\text{q}}, g_{\text{c}}$, and $g_{\text{v}}$ from Lemma 5, 6, and 7, respectively. We now show that the (modified) Transformer network $\overline{g} = g_{\text{v}} \circ g_{\text{c}} \circ g_{\text{q}}$ approximates the underlying peicewise constant function $\overline{f} \in \overline{\mathcal{F}}_{\text{PE}}$ over all points in its support except for a set of of measure $O(\delta^d)$.

Consider a point $\boldsymbol{X} \in \mathbb{S}_{\boldsymbol{L}} \subset [0,1]^{d \times n}$, where $\boldsymbol{L} \in \widetilde{\mathbb{G}}_{\delta}$. By Lemma 5, we have that $g_{\text{q}}(\boldsymbol{X}) = \boldsymbol{L}$. Thus, it follows from Lemmas 6 and 7 that

$$g_{\text{v}} \circ g_{\text{c}} \circ g_{\text{q}}(\boldsymbol{X}) = g_{\text{v}} \circ g_{\text{c}}(\boldsymbol{L}) = \begin{bmatrix} g_{\text{v}}^{\text{tkn}}(g_{\text{c}}(\boldsymbol{L})_{.,1}) & g_{\text{v}}^{\text{tkn}}(g_{\text{c}}(\boldsymbol{L})_{.,2}) & \cdots & g_{\text{v}}^{\text{tkn}}(g_{\text{c}}(\boldsymbol{L})_{.,n}) \end{bmatrix} = \boldsymbol{A}_{\boldsymbol{L}}.$$

On the other hand, any point $\boldsymbol{X} \in \bigcup_{\boldsymbol{L} \in \mathbb{G}_{\delta} \setminus \widetilde{\mathbb{G}}_{\delta}} \mathbb{S}_{\boldsymbol{L}} \cup (\mathbb{R}^{d \times n} \setminus [0,1]^{d \times n})$ is mapped by $g_{\text{q}}$ to $\boldsymbol{L} \in \mathbb{G}_{\delta}^{+} \setminus \widetilde{\mathbb{G}}_{\delta}$; as a result, we get $g_{\text{v}} \circ g_{\text{c}} \circ g_{\text{q}}(\boldsymbol{X}) = g_{\text{v}} \circ g_{\text{c}}(\boldsymbol{L}) = \boldsymbol{0}$.

Therefore, we have $\overline{g}(\boldsymbol{X}) = g_{\text{v}} \circ g_{\text{c}} \circ g_{\text{q}}(\boldsymbol{X}) = \boldsymbol{A}_{\boldsymbol{L}} = \overline{f}(\boldsymbol{X})$ for $\boldsymbol{X} \in \bigcup_{\boldsymbol{L} \in \widetilde{\mathbb{G}}_{\delta}} \mathbb{S}_{\boldsymbol{L}}$, and $\boldsymbol{0}$ everywhere else. Recall that $\overline{f}$ has its compact support in $[0,1]^d$, thus bounded; i.e., there exists $B \geq 0$ such that $\|\overline{f}(\boldsymbol{X})\|_p \leq B$. The modified Transformer network $\overline{g}$ takes the same value as $\overline{f}$ on all points in $[0,1]^d$ except for a set $\bigcup_{\boldsymbol{L} \in \mathbb{G}_{\delta} \setminus \widetilde{\mathbb{G}}_{\delta}} \mathbb{S}_{\boldsymbol{L}}$ that has measure $O(\delta^d)$. This implies that $\mathsf{d}_p(\overline{f}, \overline{g}) \leq (B^p \delta^d)^{1/p} = O(\delta^{d/p})$.

## B.4   PROOF OF LEMMA 5

The proof strategy is simple; using $\frac{1}{\delta} + 1$ token-wise feed-forward layers, we implement the quantization function $g_{\text{q}}^{\text{ent}}$ that works on the first row of the input. Then stack another $\frac{1}{\delta} + 1$ layers that quantizes the second row, and so on.

Given input $\boldsymbol{X}$, we first start by clipping $\boldsymbol{X}_{1,:}$ in the set $(-\infty, 0) \cup [1, +\infty)$ and mapping the intervals to $-\delta^{-nd}$. This can be done by the following layer:

$$\boldsymbol{Z} \mapsto \boldsymbol{Z} + \boldsymbol{e}^{(1)} \phi((\boldsymbol{e}^{(1)})^T \boldsymbol{Z}), \ \ \phi(t) = \begin{cases} -t - \delta^{-nd} & \text{if } t < 0 \text{ or } t \geq 1, \\ 0 & \text{otherwise.} \end{cases}$$

Next, add $1/\delta$ layers of the following form, for $k = 0, \delta, \dots, 1 - \delta$.

$$\boldsymbol{Z} \mapsto \boldsymbol{Z} + \boldsymbol{e}^{(1)} \phi((\boldsymbol{e}^{(1)})^T \boldsymbol{Z} - k\delta \mathbf{1}_n^T), \ \ \phi(t) = \begin{cases} 0 & t < 0 \text{ or } t \geq \delta \\ -t & 0 \leq t < \delta. \end{cases}$$

Each layer quantizes $\boldsymbol{X}_{1,:}$ in $[k\delta, k\delta + \delta)$ to $k\delta$, without modifying other intervals.

Note that both $\phi$'s used in this construction are piecewise linear functions with three pieces, and at least one of them are constant. Thus, both $\phi$'s are in $\Phi$. We can repeat the same thing for the other rows, and at the end we will get a map from $\mathbb{R}^{d \times n}$ to $\mathbb{G}_{\delta}^{+}$.

## B.5   PROOF OF LEMMA 6

**Selective shift operation.**   Before starting the proof, we first describe the key component of our proof, which we refer to the *selective shift operation*. Consider the following function, which can be expressed with a multiplicative attention head, with head size $m = 1$ and hardmax $\sigma_{\text{H}}$:

$$\psi(\boldsymbol{Z}; b_Q) = \boldsymbol{e}^{(1)} \boldsymbol{u}^T \boldsymbol{Z} \sigma_{\text{H}}[(\boldsymbol{u}^T \boldsymbol{Z})^T (\boldsymbol{u}^T \boldsymbol{Z} - b_Q \mathbf{1}_n^T)]$$

where $\boldsymbol{u} \in \mathbb{R}^d$ is a vector that we will choose later, and $\boldsymbol{e}^{(1)} = (1, 0, 0, \dots, 0) \in \mathbb{R}^d$ is the standard basis vector.

To see what this function computes, first consider the $j$-th column of the attention score matrix: $(\boldsymbol{u}^T \boldsymbol{Z})^T (\boldsymbol{u}^T \boldsymbol{Z}_{:,j} - b_Q)$. Note that, if $\boldsymbol{u}^T \boldsymbol{Z}_{:,j} > b_Q$, $\sigma_{\mathrm{H}}$ will calculate $\arg\max$ of $\boldsymbol{u}^T \boldsymbol{Z}$, whereas if $\boldsymbol{u}^T \boldsymbol{Z}_{:,j} < b_Q$, it will calculate $\arg\min$. Therefore, the $(1, j)$-th entry of $\psi(\boldsymbol{Z}; b_Q) \in \mathbb{R}^{d \times n}$ can be written as

$$\psi(\boldsymbol{Z}; b_Q)_{1,j} = \boldsymbol{u}^T \boldsymbol{Z} \sigma_{\mathrm{H}}[(\boldsymbol{u}^T \boldsymbol{Z})^T (\boldsymbol{u}^T \boldsymbol{Z}_{:,j} - b_Q)] = \begin{cases} \max_k \boldsymbol{u}^T \boldsymbol{Z}_{:,k} & \text{if } \boldsymbol{u}^T \boldsymbol{Z}_{:,j} > b_Q, \\ \min_k \boldsymbol{u}^T \boldsymbol{Z}_{:,k} & \text{if } \boldsymbol{u}^T \boldsymbol{Z}_{:,j} < b_Q, \end{cases}$$

for $j \in [n]$. Note that due to $\boldsymbol{e}^{(1)}$, all rows of $\psi(\boldsymbol{Z}; b_Q)$ except the first row are zero. From this observation, one can define a function parametrized by $b_Q$ and $b'_Q$, where $b_Q < b'_Q$, which consists of two attention heads:

$$\Psi(\boldsymbol{Z}; b_Q, b'_Q) := \psi(\boldsymbol{Z}; b_Q) - \psi(\boldsymbol{Z}; b'_Q),$$

$$\Psi(\boldsymbol{Z}; b_Q, b'_Q)_{1,j} = \begin{cases} \max_k \boldsymbol{u}^T \boldsymbol{Z}_{:,k} - \min_k \boldsymbol{u}^T \boldsymbol{Z}_{:,k} & \text{if } b_Q < \boldsymbol{u}^T \boldsymbol{Z}_{:,j} < b'_Q, \\ 0 & \text{if } \boldsymbol{u}^T \boldsymbol{Z}_{:,j} < b_Q \text{ or } \boldsymbol{u}^T \boldsymbol{Z}_{:,j} > b'_Q. \end{cases}$$

What this means is that, if we define an attention layer of the form $\boldsymbol{Z} \mapsto \boldsymbol{Z} + \Psi(\boldsymbol{Z}; b_Q, b'_Q)$, then any column $\boldsymbol{Z}_{:,j}$ satisfying $\boldsymbol{u}^T \boldsymbol{Z}_{:,j} \in (b_Q, b'_Q)$ is shifted up in its first coordinate $\boldsymbol{Z}_{1,j}$ by $\max_k \boldsymbol{u}^T \boldsymbol{Z}_{:,k} - \min_k \boldsymbol{u}^T \boldsymbol{Z}_{:,k}$, **while all the other coordinates stay untouched**. We call this the selective shift operation, because we can choose $b_Q$ and $b'_Q$ to selectively shift certain entries of the input.

**Bijective column id mapping.** Recall that the input to this step is from the range of $g_{\mathrm{q}}$ (Lemma 5), which is $\mathbb{G}_\delta^+ = \{-\delta^{-nd}, 0, \delta, \dots, 1 - \delta\}^{d \times n}$. Now consider $\boldsymbol{L} \in \mathbb{G}_\delta^+$ and $\boldsymbol{u} = (1, \delta^{-1}, \delta^{-2}, \dots, \delta^{-d+1})$.

For any $j \in [n]$, it is easy to check two following facts:

1. If $\boldsymbol{L}_{i,j} \neq -\delta^{-nd}$ for all $i \in [d]$, i.e., $\boldsymbol{L}_{:,j} \in \{0, \delta, \dots, 1 - \delta\}^d$, then $\boldsymbol{u}^T \boldsymbol{L}_{:,j} \in [0 : \delta : \delta^{-d+1} - \delta]$, and the map $\boldsymbol{L}_{:,j} \mapsto \boldsymbol{u}^T \boldsymbol{L}_{:,j}$ from $\{0, \delta, \dots, 1 - \delta\}^d$ to $[0 : \delta : \delta^{-d+1} - \delta]$ is a bijection.

2. If there exists $i \in [d]$ such that $\boldsymbol{L}_{i,j} = -\delta^{-nd}$, then $\boldsymbol{u}^T \boldsymbol{L}_{:,j} \leq -\delta^{-nd} + \delta^{-d+1} - 1 < 0$.

Therefore, one can say that $\boldsymbol{u}^T \boldsymbol{L}_{:,j}$ gives the "column id" for each possible value of $\boldsymbol{L}_{:,j} \in \{0, \delta, \dots, 1 - \delta\}^d$.

The rough idea of the construction is to apply the selective shift operation to each column id, by setting $\boldsymbol{u}$ in the definition of $\Psi(\cdot)$ to be $(1, \delta^{-1}, \delta^{-2}, \dots, \delta^{-d+1})$ and choosing $b_Q = l - \delta/2$ and $b'_Q = l + \delta/2$ for each $l \in [0 : \delta : \delta^{-d+1} - \delta]$. More concretely, we stack $(1/\delta)^d$ attention layers, with attention parts $\delta^{-d} \Psi(\cdot; l - \delta/2, l + \delta/2)$ for each $l \in [0 : \delta : \delta^{-d+1} - \delta]$, in increasing order of $l$. After that, we add an extra single-head attention layer with attention part $\delta^{-(n+1)d} \psi(\cdot; 0)$.

We now divide possible input values $\boldsymbol{L} \in \mathbb{G}_\delta^+$ into three disjoint categories, and show how these layers change the input values at the end of all the layers. Recall the hierarchy $\widetilde{\mathbb{G}}_\delta \subset \mathbb{G}_\delta \subset \mathbb{G}_\delta^+$. The categories are defined as follows:

1. $\boldsymbol{L} \in \widetilde{\mathbb{G}}_\delta$. All entries are between $0$ and $1 - \delta$, and all columns are unique.

2. $\boldsymbol{L} \in \mathbb{G}_\delta \setminus \widetilde{\mathbb{G}}_\delta$. All entries are between $0$ and $1 - \delta$, but there are duplicate columns.

3. $\boldsymbol{L} \in \mathbb{G}_\delta^+ \setminus \mathbb{G}_\delta$. The point has at least one entry that equals to $-\delta^{-nd}$.

### B.5.1 CATEGORY 1

In Category 1, we have $\boldsymbol{L} \in \widetilde{\mathbb{G}}_\delta$. Let $l_j := \boldsymbol{u}^T \boldsymbol{L}_{:,j}$. Due to permutation equivariance, we can assume without loss of generality that $l_j$'s are in increasing order: $l_1 < l_2 < \cdots < l_n$. The first $(1/\delta)^d$ layers sweep the set $[0 : \delta : \delta^{-d+1} - \delta]$ and apply selective shift operation on each element in the set. This means that selective shift operation will be applied to $l_1$ first, then $l_2$, and then $l_3$, and so on, regardless of the specific values of $l_j$'s.

**First shift operation.** In the first selective shift operation, the $(1,1)$-th entry of $\boldsymbol{L}$ ($L_{1,1}$) is shifted by the operation, while the other entries are left untouched. The updated value $\widetilde{L}_{1,1}$ is

$$\widetilde{L}_{1,1} = L_{1,1} + \delta^{-d}(\max_k \boldsymbol{u}^T \boldsymbol{L}_{:,k} - \min_k \boldsymbol{u}^T \boldsymbol{L}_{:,k}) = L_{1,1} + \delta^{-d}(l_n - l_1).$$

Therefore, after the operation, the output of the layer is $\begin{bmatrix} \widetilde{\boldsymbol{L}}_{:,1} & \boldsymbol{L}_{:,2} & \ldots & \boldsymbol{L}_{:,n} \end{bmatrix}$, and the new value of the first column $\widetilde{\boldsymbol{L}}_{:,1}$ results in

$$\boldsymbol{u}^T \widetilde{\boldsymbol{L}}_{:,1} = \widetilde{L}_{1,1} + \sum_{i=2}^{d} \delta^{-i+1} L_{i,1} = L_{1,1} + \delta^{-d}(l_n - l_1) + \sum_{i=2}^{d} \delta^{-i+1} L_{i,1} = l_1 + \delta^{-d}(l_n - l_1).$$

Let us denote the updated "column id" $\boldsymbol{u}^T \widetilde{\boldsymbol{L}}_{:,1}$ as $\widetilde{l}_1$. We can show that $l_n < \widetilde{l}_1$, because

$$\widetilde{l}_1 := l_1 + \delta^{-d}(l_n - l_1) \geq 0 + \delta^{-d} \cdot \delta = \delta^{-d+1} > l_n.$$

Therefore, after updating,

$$\max \boldsymbol{u}^T \begin{bmatrix} \widetilde{\boldsymbol{L}}_{:,1} & \boldsymbol{L}_{:,2} & \ldots & \boldsymbol{L}_{:,n} \end{bmatrix} = \max\{\widetilde{l}_1, l_2, \ldots, l_n\} = \widetilde{l}_1,$$

and the new minimum is $l_2$.

**Second shift operation.** The second selective shift operation is applied to $l_2$, by which only one entry $L_{1,2}$ will be shifted. The updated value $\widetilde{L}_{1,2}$ is

$$\widetilde{L}_{1,2} = L_{1,2} + \delta^{-d}(\widetilde{l}_1 - l_2) = L_{1,2} + \delta^{-d}(l_1 - l_2) + \delta^{-2d}(l_n - l_1).$$

After updating, the new inner product of $\boldsymbol{u}$ and $\widetilde{\boldsymbol{L}}_{:,2}$ results in

$$\widetilde{l}_2 := \boldsymbol{u}^T \widetilde{\boldsymbol{L}}_{:,2} = l_2 + \delta^{-d}(l_1 - l_2) + \delta^{-2d}(l_n - l_1).$$

We can show that $\widetilde{l}_1 < \widetilde{l}_2$, because

$$l_1 + \delta^{-d}(l_n - l_1) < l_2 + \delta^{-d}(l_1 - l_2) + \delta^{-2d}(l_n - l_1)$$
$$\Leftrightarrow (\delta^{-d} - 1)(l_2 - l_1) < \delta^{-d}(\delta^{-d} - 1)(l_n - l_1),$$

and the last inequality is true because $\delta^{-d} > 1$ and $l_n > l_2$. Since we have $\widetilde{l}_1 < \widetilde{l}_2$, and the new maximum in $\boldsymbol{u}^T \begin{bmatrix} \widetilde{\boldsymbol{L}}_{:,1} & \widetilde{\boldsymbol{L}}_{:,2} & \boldsymbol{L}_{:,3} & \ldots & \boldsymbol{L}_{:,n} \end{bmatrix}$ is now $\widetilde{l}_2$, and the new minimum is $l_3$.

**Repeating the process.** More generally, we can repeat this process, and show that the $j$-th shift operation shifts $L_{1,j}$ by $\delta^{-d}(\widetilde{l}_{j-1} - l_j)$, and results in the new column id

$$\widetilde{l}_j := \boldsymbol{u}^T \widetilde{\boldsymbol{L}}_{:,j} = l_j + \sum_{k=1}^{j-1} \delta^{-kd}(l_{j-k} - l_{j-k+1}) + \delta^{-jd}(l_n - l_1).$$

In the general case, $\widetilde{l}_{j-1} < \widetilde{l}_j$ holds $j = [2:n]$, because

$$\widetilde{l}_{j-1} = l_{j-1} + \sum_{k=2}^{j-1} \delta^{-kd+d}(l_{j-k} - l_{j-k+1}) + \delta^{-(j-1)d}(l_n - l_1)$$

$$< \widetilde{l}_j = l_j + \sum_{k=1}^{j-1} \delta^{-kd}(l_{j-k} - l_{j-k+1}) + \delta^{-jd}(l_n - l_1)$$

$$\Leftrightarrow \sum_{k=1}^{j-1} \delta^{-kd+d}(\delta^{-d} - 1)(l_{j-k+1} - l_{j-k}) < \delta^{-(j-1)d}(\delta^{-d} - 1)(l_n - l_1),$$

and the last inequality holds because

$$\delta^{-(j-1)d}(l_n - l_1) > \delta^{-(j-1)d} \sum_{k=1}^{j-1}(l_{j-k+1} - l_{j-k}) > \sum_{k=1}^{j-1} \delta^{-kd+d}(l_{j-k+1} - l_{j-k}).$$

Therefore, after the $j$-th selective shift operation, $\widetilde{l}_j$ is the new maximum among $\{\widetilde{l}_1, \ldots, \widetilde{l}_j, l_{j+1}, \ldots, l_n\}$ and $l_{j+1}$ is the new minimum, which makes us possible to continue the process until the $n$-th operation.

**After $n$ shift operations.** As a result, after the whole sweep from $0$ to $\delta^{-d+1} - \delta$ by the first $(1/\delta)^d$ layers, a total of $n$ shift operations are applied, and the input $\boldsymbol{L}$ is mapped to a new point $\widetilde{\boldsymbol{L}}$, where $\boldsymbol{u}^T \widetilde{\boldsymbol{L}} = \begin{bmatrix} \widetilde{l}_1 & \widetilde{l}_2 & \dots & \widetilde{l}_n \end{bmatrix}$ and $\widetilde{l}_1 < \widetilde{l}_2 < \cdots < \widetilde{l}_n$.

We can now prove the following technical lemma, whose proof is deferred to Appendix B.5.4:

**Lemma 10.** *After $n$ shift operations, $\widetilde{l}_n = \boldsymbol{u}^T \widetilde{\boldsymbol{L}}_{:,n}$ satisfies the following bounds:*

$$\delta^{-(n-1)d+1}(\delta^{-d} - 1) \le \widetilde{l}_n \le \delta^{-nd+1}(\delta^{-d} - 1) - \delta(\delta^{-d} - 1)^2.$$

*Also, the map from $\begin{bmatrix} l_1 & l_2 & \cdots & l_n \end{bmatrix} \in [0 : \delta : \delta^{-d+1} - \delta]$ (where $l_1 < l_2 < \cdots < l_n$) to $\widetilde{l}_n$ is one-to-one.*

**Global shifting by the last layer.** As mentioned earlier, after this sweep, there is another attention layer with attention part $\delta^{-(n+1)d} \psi(\cdot; 0)$. Since $0 < \widetilde{l}_1 < \cdots < \widetilde{l}_n$, what it does to $\widetilde{\boldsymbol{L}}$ is that it adds $\delta^{-(n+1)d} \max_k \boldsymbol{u}^T \widetilde{\boldsymbol{L}}_{:,k} = \delta^{-(n+1)d} \widetilde{l}_n$ to each entry in the first row of $\widetilde{\boldsymbol{L}}$. The output of this layer is defined to be the function $g_{\mathrm{c}}(\boldsymbol{L})$.

Now, in summary, for any $\boldsymbol{L} \in \widetilde{\mathbb{G}}_\delta$, $i \in [d]$, and $j \in [n]$, we have

$$g_{\mathrm{c}}(\boldsymbol{L})_{i,j} = \begin{cases} L_{1,j} + \sum_{k=1}^{j-1} \delta^{-kd}(l_{j-k} - l_{j-k+1}) + \delta^{-jd}(l_n - l_1) + \delta^{-(n+1)d} \widetilde{l}_n & \text{if } i = 1, \\ L_{i,j} & \text{if } i \in [2, d], \end{cases}$$

and for any $\boldsymbol{L} \in \widetilde{\mathbb{G}}_\delta$ and $j \in [n]$,

$$\boldsymbol{u}^T g_{\mathrm{c}}(\boldsymbol{L})_{:,j} = \widetilde{l}_j + \delta^{-(n+1)d} \widetilde{l}_n.$$

**Checking Properties 6.1 and 6.2.** Given this result so far, it is now left to check if the constructed network is really a permutation equivariant contextual mapping, i.e., if it satisfies Properties 6.1 and 6.2 in Lemma 6.

First, for any $\boldsymbol{L} \in \widetilde{\mathbb{G}}_\delta$, Property 6.1 holds because we already know $\widetilde{l}_1 < \widetilde{l}_2 < \cdots < \widetilde{l}_n$, so they are all distinct. As for Property 6.2, note that the upper bound on $\widetilde{l}_n$ from Lemma 10 also holds for other $\widetilde{l}_j$'s, so

$$\boldsymbol{u}^T g_{\mathrm{c}}(\boldsymbol{L})_{:,j} \in [\delta^{-(n+1)d} \widetilde{l}_n, \delta^{-(n+1)d} \widetilde{l}_n + \delta^{-(n+1)d+1}),$$

for all $j \in [n]$. Now, from Lemma 10, two $\boldsymbol{L}, \boldsymbol{L}' \in \widetilde{\mathbb{G}}_\delta$ (that are not permutations of each other) map to different $\widetilde{l}_n$ and $\widetilde{l}'_n$, and they differ at least by $\delta$. This means that two intervals $[\delta^{-(n+1)d} \widetilde{l}_n, \delta^{-(n+1)d} \widetilde{l}_n + \delta^{-(n+1)d+1})$ and $[\delta^{-(n+1)d} \widetilde{l}'_n, \delta^{-(n+1)d} \widetilde{l}'_n + \delta^{-(n+1)d+1})$ are guaranteed to be disjoint, so the entries of $\boldsymbol{u}^T g_{\mathrm{c}}(\boldsymbol{L})$ and $\boldsymbol{u}^T g_{\mathrm{c}}(\boldsymbol{L}')$ are all distinct. This proves Property 6.2.

Therefore, we finished showing that the map $g_{\mathrm{c}}(\cdot)$ we constructed using $(1/\delta)^d + 1$ attention layers implements a permutation equivariant contextual mapping on $\widetilde{\mathbb{G}}_\delta$.

**Checking Property 6.3.** It is now left to check if the map $g_{\mathrm{c}}$ satisfies the other properties. At this point, we can check Property 6.3. From $\boldsymbol{u}^T g_{\mathrm{c}}(\boldsymbol{L})_{:,j} \in [\delta^{-(n+1)d} \widetilde{l}_n, \delta^{-(n+1)d} \widetilde{l}_n + \delta^{-(n+1)d+1})$ and Lemma 10, we can show that for any $\boldsymbol{L} \in \widetilde{\mathbb{G}}_\delta$, we have

$$\delta^{-2nd+1}(\delta^{-d} - 1) \le \boldsymbol{u}^T g_{\mathrm{c}}(\boldsymbol{L})_{:,j} < \delta^{-(n+1)d}(\delta^{-nd+1}(\delta^{-d} - 1) - \delta(\delta^{-d} - 1)^2) + \delta^{-(n+1)d+1}$$
$$\le \delta^{-(2n+1)d+1}(\delta^{-d} - 1),$$

where we used $\delta^{-1} \ge 2$. This proves that all $\boldsymbol{u}^T g_{\mathrm{c}}(\boldsymbol{L})_{:,j}$ are between $t_l = \delta^{-2nd+1}(\delta^{-d} - 1)$ and $t_r = \delta^{-(2n+1)d+1}(\delta^{-d} - 1)$. For the remaining input points $\boldsymbol{L} \in \mathbb{G}_\delta^+ \setminus \widetilde{\mathbb{G}}_\delta$, we will check that $\boldsymbol{u}^T g_{\mathrm{c}}(\boldsymbol{L})_{:,j}$ is outside the interval $[t_l, t_r]$ (Property 6.4).

### B.5.2   CATEGORY 2

In Category 2, we have $\boldsymbol{L} \in \mathbb{G}_\delta \setminus \widetilde{\mathbb{G}}_\delta$. Here, all entries are between $0$ and $1 - \delta$, but there are duplicate columns. Again, let $l_j := \boldsymbol{u}^T \boldsymbol{L}_{:,j}$, and assume without loss of generality that $l_1 \le l_2 \le \cdots \le l_n$.

For the input $L$ in Category 2, there exist some $j, j' \in [n]$, $j \neq j'$, such that $l_j = l_{j'}$. This means that when the input passes through the attention layer $\delta^{-d}\Psi(\cdot; l_j - \delta/2, l_j + \delta/2)$, the selective shift operation for $l_j$ is applied to both $j$-th and $j'$-th columns; the two columns are coupled together. More generally, suppose we have $n' < n$ distinct columns.

**If $n' = 1$.** In the extreme case of $n' = 1$, we have $\max_j l_j = \min_j l_j$, so the selective shift operation applied at $l_j$ does not shift the entry at all; therefore, at the end of the first $(1/\delta)^d$ attention layers, $\widetilde{L} = L$.

**If $1 < n' \leq n - 1$.** When $1 < n' \leq n - 1$, let the $n'$ distinct values of $l_j$'s be $l'_1, \ldots, l'_{n'}$. The shift operation is applied $n'$ times, to $l'_1, \ldots, l'_{n'}$, and shifts one or more entries at a time. After the first $(1/\delta)^d$ layers, the output $\widetilde{L}$ has $n'$ distinct $\widetilde{l}_j = \boldsymbol{u}^T \widetilde{L}_{:,j}$, $0 \leq \widetilde{l}_1 \leq \widetilde{l}_2 \leq \cdots \leq \widetilde{l}_n$, whose distinct values are the same as the numbers we get when we apply shift operations to a length-$n'$ sequence $[l'_1 \quad \cdots \quad l'_{n'}]$. Then, applying the same calculations from Category 1 shows that

$$\widetilde{l}_n = \boldsymbol{u}^T \widetilde{L}_{:,n} = l'_{n'} + \sum_{k=1}^{n'-1} \delta^{-kd}(l'_{n'-k} - l'_{n'-k+1}) + \delta^{-n'd}(l'_{n'} - l'_1),$$

and it follows from the upper bound in Lemma 10 that

$$\widetilde{l}_n \leq \delta^{-n'd+1}(\delta^{-d} - 1) - \delta(\delta^{-d} - 1)^2 < \delta^{-(n-1)d+1}(\delta^{-d} - 1).$$

Note that the RHS matches the lower bound in Lemma 10. This implies that the value of $\widetilde{l}_n$ calculated from the input $L \in \mathbb{G}_\delta \setminus \widetilde{\mathbb{G}}_\delta$ (Category 2) is always strictly less (by at least $\delta$) than that calculated from $L \in \widetilde{\mathbb{G}}_\delta$ (Category 1).

**Checking Property 6.4.** After the global shifting by the last layer with attention part $\delta^{-(n+1)d}\psi(\cdot; 0)$, we get the output $g_c(L)$ which satisfies

$$\boldsymbol{u}^T g_c(L)_{:,j} = \widetilde{l}_j + \delta^{-(n+1)d}\widetilde{l}_n \leq (\delta^{-(n+1)d} + 1)(\delta^{-(n-1)d+1}(\delta^{-d} - 1) - \delta(\delta^{-d} - 1)^2)$$
$$< \delta^{-2nd+1}(\delta^{-d} - 1) =: t_l.$$

where the RHS is a lower bound on possible values of $\boldsymbol{u}^T g_c(L)_{:,j}$ for $L \in \widetilde{\mathbb{G}}_\delta$ (Category 1). This means that the entries of $\boldsymbol{u}^T g_c(L)$ for Category 2 are outside $[t_l, t_r]$, which satisfies Property 6.4.

### B.5.3 CATEGORY 3

In Category 3, we have $L \in \mathbb{G}_\delta^+ \setminus \mathbb{G}_\delta$; the point $L$ has at least one entry that equals to $-\delta^{-nd}$. Let $l_j := \boldsymbol{u}^T L_{:,j}$, and recall that whenever a column $L_{:,j}$ has an entry that equals to $-\delta^{-nd}$, we have $l_j = \boldsymbol{u}^T L_{:,j} \leq -\delta^{-nd} + \delta^{-d+1} - 1 < 0$. Assume without loss of generality that $l_1 \leq l_2 \leq \cdots \leq l_n$.

Recall that the selective shift operation is applied to each element of $[0 : \delta : \delta^{-d+1} - \delta]$, not to negative values. In case of Category 3, we have $\min_k \boldsymbol{u}^T L_{:,k} = l_1 < 0$, and $l_1$ never gets shifted upwards, so it remains as the minimum for the whole time.

**If all $l_j$'s are negative.** In case where all $l_j$'s are negative, selective shift operation never changes the input $L$, so we get $\widetilde{L} = L$. Since we have $\boldsymbol{u}^T \widetilde{L} < \boldsymbol{0}_n^T$ (entry-wise), the last layer with attention part $\delta^{-(n+1)d}\psi(\cdot; 0)$ adds $\delta^{-(n+1)d}\min_k \boldsymbol{u}^T \widetilde{L}_{:,k} < 0$ to each entry in the first row of $\widetilde{L}$, further pushing it to the negative side. Therefore, the final output $g_c(L)$ satisfies $\boldsymbol{u}^T g_c(L) < \boldsymbol{0}_n^T < t_l \boldsymbol{1}_n^T$.

**If not all $l_j$'s are negative.** Now consider the case where at least one $l_j$ is positive. Let $i$ be the index that satisfies $l_{i-1} < 0 \leq l_i$. Then, selective shift operation does not affect $l_1, \ldots, l_{i-1}$, and then it shifts $l_i$ by

$$\delta^{-d}(\max_k \boldsymbol{u}^T L_{:,k} - \min_k \boldsymbol{u}^T L_{:,k}) = \delta^{-d}(l_n - l_1) \geq \delta^{-d}(0 + \delta^{-nd} - \delta^{-d+1} + 1) \geq \delta^{-(n+1)d+1},$$

where we used $\delta^{-1} \geq 2$ at the last inequality. The next shift operations shift $l_{i+1}, \ldots, l_n$ by even larger amount, so at the end of the first $(1/\delta)^d$ layers, we have $\delta^{-(n+1)d+1} \leq \widetilde{l}_i \leq \cdots \leq \widetilde{l}_n$, while $\widetilde{l}_j = l_j < 0$ for $j \in [i-1]$.

**Shifts by the last layer.** Here, the last layer with attention part $\delta^{-(n+1)d}\psi(\cdot;0)$ acts differently for negative and positive $\widetilde{l}_j$'s. For negative $\widetilde{l}_j$'s, it adds $\delta^{-(n+1)d}\min_k \widetilde{l}_k = \delta^{-(n+1)d}l_1 < 0$ to $\widetilde{l}_1, \ldots, \widetilde{l}_{i-1}$, pushing them further to the negative side. For positive $\widetilde{l}_j$'s, the layer adds $\delta^{-(n+1)d}\max_k \widetilde{l}_k = \delta^{-(n+1)d}\widetilde{l}_n \geq \delta^{-(2n+2)d+1}$ to $\widetilde{l}_i, \ldots, \widetilde{l}_n$, so that they are all greater than or equal to $\delta^{-(2n+2)d+1}$. Note that $\delta^{-(2n+2)d+1} > t_r$.

**Checking Property 6.4.** Therefore, in both cases, we can see that the final output $g_c(\boldsymbol{L})$ satisfies $\boldsymbol{u}^T g_c(\boldsymbol{L})_{:,j} \notin [t_l, t_r]$, for all $j \in [n]$. This completes the verification of Property 6.4.

### B.5.4 PROOF OF LEMMA 10

Proof of lower and upper bounds on $\widetilde{l}_n$ are straightforward:

$$
\begin{aligned}
\widetilde{l}_n &:= l_n + \sum_{k=1}^{n-1} \delta^{-kd}(l_{n-k} - l_{n-k+1}) + \delta^{-nd}(l_n - l_1) \\
&\geq \delta^{-(n-1)d} \sum_{k=1}^{n-1}(l_{n-k} - l_{n-k+1}) + \delta^{-nd}(l_n - l_1) = (\delta^{-nd} - \delta^{-(n-1)d})(l_n - l_1) \\
&\geq \delta^{-(n-1)d+1}(\delta^{-d} - 1), \\
\widetilde{l}_n &\leq l_n + \delta^{-d}(l_1 - l_n) + \delta^{-nd}(l_n - l_1) \leq \delta^{-d+1} - \delta + (\delta^{-nd} - \delta^{-d})(\delta^{-d+1} - \delta) \\
&= \delta^{-nd+1}(\delta^{-d} - 1) - \delta(\delta^{-2d} - 2\delta^{-d} + 1) = \delta^{-nd+1}(\delta^{-d} - 1) - \delta(\delta^{-d} - 1)^2.
\end{aligned}
$$

For one-to-one property of the map, consider $[l_1 \quad l_2 \quad \cdots \quad l_n]$ and $[l'_1 \quad l'_2 \quad \cdots \quad l'_n]$ with increasing entries, which are mapped to $\widetilde{l}_n$ and $\widetilde{l}'_n$, respectively. Suppose $\widetilde{l}_n = \widetilde{l}'_n$. By definition,

$$
\begin{aligned}
\widetilde{l}_n - \widetilde{l}'_n &= (l_n - l'_n) + \delta^{-d}(l_{n-1} - l_n - l'_{n-1} + l'_n) + \delta^{-2d}(l_{n-2} - l_{n-1} - l'_{n-2} + l'_{n-1}) + \ldots \\
&\quad + \delta^{-(n-1)d}(l_1 - l_2 - l'_1 + l'_2) + \delta^{-nd}(l_n - l_1 - l'_n + l'_1) = 0.
\end{aligned}
$$

Now assume for contradiction that $l_n \neq l'_n$. Then, we have $-\delta^{-d+1} + \delta \leq l_n - l'_n \leq \delta^{-d+1} - \delta$. However, the remaining terms have "coarse resolution", and they can never cancel $l_n - l'_n$ and make the sum zero, because for example, $\delta^{-d}(l_{n-1} - l_n - l'_{n-1} + l'_n)$ can only have values $0, \delta^{-d+1}, -\delta^{-d+1}, 2\delta^{-d+1}, -2\delta^{-d+1}, \ldots$. Thus, $l_n = l'_n$ must hold and the first term must be zero.

Similarly, assume that $l_{n-1} \neq l'_{n-1}$. Then, the second term is in the interval $[-\delta^{-2d+1} + \delta^{-d+1}, \delta^{-2d+1} - \delta^{-d+1}]$. Again, the remaining terms cannot cancel the second term, hence $l_{n-1} = l'_{n-1}$ must hold. We can proceed this way, and show that $l_j = l'_j$ must hold for all $j \in [n]$, hence proving that the map is one-to-one.

### B.6 PROOF OF LEMMA 7

Note that $|\mathbb{G}_\delta^+| = (\frac{1}{\delta} + 1)^{dn}$, so the image of $g_c(\mathbb{G}_\delta^+)$ (from Lemma 6) has finite number of distinct real numbers. Let $M$ be the maximum over all these numbers. By construction of $g_c$, we know that $M > 0$.

To construct a function $g_v^{\text{tkn}}$ that satisfies the statement of the lemma, we first implement the second part: $g_v^{\text{tkn}}(g_c(\boldsymbol{L})_{:,j}) = \boldsymbol{0}_d$ if $\boldsymbol{L} \in \mathbb{G}_\delta^+ \setminus \widetilde{\mathbb{G}}_\delta$. Note from Lemma 6 that, for any $\boldsymbol{L} \in \widetilde{\mathbb{G}}_\delta$, we have $\boldsymbol{u}^T g_c(\boldsymbol{L})_{:,j} \in [t_l, t_r]$ for all $j$, and for any $\boldsymbol{L} \in \mathbb{G}_\delta^+ \setminus \widetilde{\mathbb{G}}_\delta$, $\boldsymbol{u}^T g_c(\boldsymbol{L})_{:,j} \notin [t_l, t_r]$ for all $j$. Using this, we add the following feed-forward layer:

$$
\boldsymbol{Z} \mapsto \boldsymbol{Z} - (M+1)\boldsymbol{1}_n \phi(\boldsymbol{u}^T \boldsymbol{Z}), \quad \phi(t) = \begin{cases} 0 & \text{if } t \in [t_l, t_r] \\ 1 & \text{if } t \notin [t_l, t_r]. \end{cases}
$$

Input to this layer is $g_c(\boldsymbol{L})$. If $\boldsymbol{L} \in \widetilde{\mathbb{G}}_\delta$, then $\phi(\boldsymbol{u}^T g_c(\boldsymbol{L})) = \boldsymbol{0}_n^T$, so the output stays the same as the input. If $\boldsymbol{L} \in \mathbb{G}_\delta^+ \setminus \widetilde{\mathbb{G}}_\delta$, then $\phi(\boldsymbol{u}^T g_c(\boldsymbol{L})) = \boldsymbol{1}_n^T$, so all the entries of the input are shifted by $-M - 1$, and become strictly negative.

Recall that by definition of $\widetilde{\mathbb{G}}_\delta$, all the entries of $g_{\mathrm{c}}(\boldsymbol{L})$ for $\boldsymbol{L} \in \widetilde{\mathbb{G}}_\delta$ are nonnegative. So the next thing to do is mapping all strictly negative entries to zero. This can be done in a similar way as Lemma 5. For $i \in [d]$, add the following layer:

$$\boldsymbol{Z} \mapsto \boldsymbol{Z} + \boldsymbol{e}^{(i)}\phi((\boldsymbol{e}^{(i)})^T \boldsymbol{Z}), \;\; \phi(t) = \begin{cases} -t & \text{if } t < 0 \\ 0 & \text{if } t \geq 0. \end{cases}$$

After these $d$ layers, the output for $\boldsymbol{L} \in \mathbb{G}_\delta^+ \setminus \widetilde{\mathbb{G}}_\delta$ is a zero matrix, while the output for $\boldsymbol{L} \in \widetilde{\mathbb{G}}_\delta$ is $g_{\mathrm{c}}(\boldsymbol{L})$.

Now, it is left to map $g_{\mathrm{c}}(\boldsymbol{L})$ to $\boldsymbol{A_L}$, for $\boldsymbol{L} \in \widetilde{\mathbb{G}}_\delta$. Up to permutation equivariance, each different context $\boldsymbol{L}$ maps to $n$ unique numbers $\boldsymbol{u}^T g_{\mathrm{c}}(\boldsymbol{L})$, which are at least $\delta$ apart from each other. The idea of value mapping is to map each unique number to the corresponding output column.

More precisely, choose any $\overline{\boldsymbol{L}} \in \widetilde{\mathbb{G}}_\delta$. For each value of $\boldsymbol{u}^T g_{\mathrm{c}}(\overline{\boldsymbol{L}})_{:,j}$, $j \in [n]$, we add one feed-forward layer

$$\boldsymbol{Z} \mapsto \boldsymbol{Z} + ((\boldsymbol{A}_{\overline{\boldsymbol{L}}})_{:,j} - g_{\mathrm{c}}(\overline{\boldsymbol{L}})_{:,j})\phi(\boldsymbol{u}^T \boldsymbol{Z} - \boldsymbol{u}^T g_{\mathrm{c}}(\overline{\boldsymbol{L}})_{:,j}\boldsymbol{1}_n^T), \;\; \phi(t) = \begin{cases} 0 & t < -\delta/2 \text{ or } t \geq \delta/2, \\ 1 & -\delta/2 \leq t < \delta/2. \end{cases}$$

If the input $\boldsymbol{Z}$ is a zero matrix, which is the case for $\boldsymbol{L} \in \mathbb{G}_\delta^+ \setminus \widetilde{\mathbb{G}}_\delta$, $\boldsymbol{u}^T \boldsymbol{Z} = \boldsymbol{0}_n^T$. Since $t_l$ is much larger than 0, activation is all zero. Thus, zero input matrix remains the same at the output.

If the input $\boldsymbol{Z}$ is $g_{\mathrm{c}}(\boldsymbol{L})$, where $\boldsymbol{L} \in \widetilde{\mathbb{G}}_\delta$ is not a permutation of $\overline{\boldsymbol{L}}$, then

$$\phi(\boldsymbol{u}^T g_{\mathrm{c}}(\boldsymbol{L}) - \boldsymbol{u}^T g_{\mathrm{c}}(\overline{\boldsymbol{L}})_{:,j}\boldsymbol{1}_n^T) = \boldsymbol{0}_n^T,$$

so $g_{\mathrm{c}}(\boldsymbol{L})$ is left untouched.

If some other $\boldsymbol{L}$ is a permutation of $\overline{\boldsymbol{L}}$, and $\boldsymbol{L}_{:,i} = \overline{\boldsymbol{L}}_{:,j}$, then

$$\phi(\boldsymbol{u}^T g_{\mathrm{c}}(\boldsymbol{L}) - \boldsymbol{u}^T g_{\mathrm{c}}(\overline{\boldsymbol{L}})_{:,j}\boldsymbol{1}_n^T) = (\boldsymbol{e}^{(i)})^T,$$

so $i$-th column of $g_{\mathrm{c}}(\boldsymbol{L})$ will turn to

$$g_{\mathrm{c}}(\boldsymbol{L})_{:,i} \mapsto g_{\mathrm{c}}(\boldsymbol{L})_{:,i} + ((\boldsymbol{A}_{\overline{\boldsymbol{L}}})_{:,j} - g_{\mathrm{c}}(\overline{\boldsymbol{L}})_{:,j}) = g_{\mathrm{c}}(\boldsymbol{L})_{:,i} + ((\boldsymbol{A_L})_{:,i} - g_{\mathrm{c}}(\boldsymbol{L})_{:,i}) = (\boldsymbol{A_L})_{:,i},$$

which is the desired output. In conclusion, this layer maps the column $g_{\mathrm{c}}(\overline{\boldsymbol{L}})_{:,j}$ to $(\boldsymbol{A}_{\overline{\boldsymbol{L}}})_{:,j}$, without affecting any other columns.

As seen above, we need one layer per each unique value of $\boldsymbol{u}^T g_{\mathrm{c}}(\boldsymbol{L})_{:,j}$ for each $\boldsymbol{L} \in \widetilde{\mathbb{G}}_\delta$. Note that there are $O(n(1/\delta)^{dn}/n!)$ such numbers, so we can use $O(n(1/\delta)^{dn}/n!)$ layers to finish our construction.

## C    Proof of Theorem 3

Proof of Theorem 3 can be done in a similar way as Theorem 2. As in the proof of Theorem 2, there are three parts: Lemma 8, Proposition 4, and Lemma 9. The statement and proof of Lemmas 8 and 9 can be done in almost the same way, this time without permutation equivariance.

For the proof of the second part, which corresponds to Proposition 4, we construct the network in a similar way. Recall that we can assume without loss of generality that $\boldsymbol{X} \in [0,1]^{d \times n}$. Choose

$$\boldsymbol{E} = \begin{bmatrix} 0 & 1 & 2 & \cdots & n-1 \\ 0 & 1 & 2 & \cdots & n-1 \\ \vdots & \vdots & \vdots & & \vdots \\ 0 & 1 & 2 & \cdots & n-1 \end{bmatrix}.$$

Then, the first column of $\boldsymbol{X} + \boldsymbol{E}$ is in $[0,1]^d$, second is in $[1,2]^d$, and so on; this means that for all rows, the coordinates are monotonically increasing. So we can use the same technique as the proof of Proposition 4 to divide the input values into cubes, quantize them to $\boldsymbol{L}$, apply contextual mapping, and then value mapping. We describe each step in the following.

## C.1 QUANTIZATION BY FEED-FORWARD LAYERS

In a similar way as Lemma 5, the goal of this step is to quantize the input in $[0,1]^d \times [1,2]^d \times \cdots \times [n-1,n]^d$ to its discrete version:

$$[0:\delta:1-\delta]^d \times [1:\delta:2-\delta]^d \times \cdots \times [n-1:\delta:n-\delta]^d.$$

This can be done by $dn/\delta$ feed-forward layers. We add $dn/\delta$ layers of the following form, for $k = 0, \delta, \ldots, n-\delta$ and $i = 1, \ldots, d$:

$$\boldsymbol{Z} \mapsto \boldsymbol{Z} + \boldsymbol{e}^{(i)}\phi((\boldsymbol{e}^{(i)})^T\boldsymbol{Z} - k\delta\mathbf{1}_n^T), \quad \phi(t) = \begin{cases} 0 & t < 0 \text{ or } t \geq \delta \\ -t & 0 \leq t < \delta. \end{cases}$$

After $dn/\delta$ layers, any input entry of $\boldsymbol{X} + \boldsymbol{E}$ in $[k\delta, k\delta+\delta)$ is quantized to $k\delta$.

## C.2 CONTEXTUAL MAPPING BY ATTENTION LAYERS

By Step 1, we quantized any input $\boldsymbol{X} + \boldsymbol{E}$ to its quantized version. We call this quantized version $\boldsymbol{L}$:

$$\boldsymbol{L} \in [0:\delta:1-\delta]^d \times [1:\delta:2-\delta]^d \times \cdots \times [n-1:\delta:n-\delta]^d.$$

As done in Lemma 6, we define $\boldsymbol{u} := (1, \delta^{-1}, \ldots, \delta^{-d+1})$ and $l_j := \boldsymbol{u}^T\boldsymbol{L}_{:,j}$, for all $j \in [n]$. Note that, because $\boldsymbol{L}_{:,j} \in [j-1:\delta:j-\delta]^d$, we have

$$(j-1)(1 + \delta^{-1} + \cdots + \delta^{-d+1}) \leq l_j \leq (j-1)(1 + \delta^{-1} + \cdots + \delta^{-d+1}) + \delta^{-d+1} - \delta,$$

and $l_1 < l_2 < \cdots < l_n$. Notice that this corresponds to the Category 1 in the proof of Lemma 6.

For simplicity of notation, let $s_j = (j-1)\sum_{k=0}^{d-1}\delta^{-k}$. We stack $n(1/\delta)^d$ attention layers, with attention parts $\delta^{-d}\Psi(\cdot; l-\delta/2, l+\delta/2)$ for each $l \in \bigcup_{j=1}^n [s_j : \delta : s_j + \delta^{-d+1} - \delta]$, in increasing order of $l$.

These $n(1/\delta)^d$ attention layers perform selective shift operations on $l_j$'s, in increasing order of $j$. As seen in Appendix B.5.1, shift operations result in $\widetilde{l}_1 < \widetilde{l}_2 < \cdots < \widetilde{l}_n$. Also, the map from $\boldsymbol{L}$ to $\widetilde{l}_n$ is one-to-one, which can be shown in the same way as Appendix B.5.4. Since the range of $l_j$'s are a bit different, we have a different upper bound on $\widetilde{l}_n$:

$$
\begin{aligned}
\widetilde{l}_n &:= l_n + \sum_{k=1}^{n-1}\delta^{-kd}(l_{n-k} - l_{n-k+1}) + \delta^{-nd}(l_n - l_1) \\
&\leq l_n + \delta^{-d}(l_1 - l_n) + \delta^{-nd}(l_n - l_1) \leq s_n + \delta^{-d+1} - \delta + (\delta^{-nd} - \delta^{-d})(s_n + \delta^{-d+1} - \delta) \\
&= (\delta^{-nd} - \delta^{-d} + 1)\left((n-1)\frac{\delta^{-d} - 1}{\delta^{-1} - 1} + \delta^{-d+1} - \delta\right) \\
&\leq (\delta^{-nd} - \delta^{-d} + 1)(\delta^{-d} - 1)(n - 1 + \delta) < n\delta^{-(n+1)d}.
\end{aligned}
$$

Finally, we add an extra single-head attention layer with attention part $n\delta^{-(n+1)d-1}\psi(\cdot; 0)$. We define the output of this layer as $g_c(\boldsymbol{L})$. In a similar way as Appendix B.5.1, this layer shifts all the layers by $n\delta^{-(n+1)d-1}\widetilde{l}_n$, thus making the intervals corresponding to different values of $\widetilde{l}_n$ disjoint from each other. This ensures that different contexts $\boldsymbol{L}$ are mapped to distinct numbers in $\boldsymbol{u}^T g_c(\boldsymbol{L})$, thus implementing a contextual mapping.

## C.3 FUNCTION VALUE MAPPING BY FEED-FORWARD LAYERS

Now, it is left to map $g_c(\boldsymbol{L})$ to the desired output. As seen in the last step, each different context $\boldsymbol{L}$ maps to $n$ unique numbers $\boldsymbol{u}^T g_c(\boldsymbol{L})$, which are at least $\delta$ apart from each other. The value mapping step can be done in a similar way as Lemma 7. The construction now requires $O(n(1/\delta)^{dn})$ layers because there is no permutation equivariance.

| Architecture | Average Attention | BProj | SepConv | Transformer |
|---|---|---|---|---|
| # params | 88.3M | 90M | 102.5M | 110M |
| Masked LM accuracy (%) | 28 | 59 | 60 | 63 |
| MNLI accuracy (%) | 66 | 72.3 | 73 | 78.2 |

Table 1: Performance of bi-linear projection and separable convolution layers on masked LM pre-training task and MNLI. Note that we expect these computationally cheaper models to have lower performance than the expensive Transformers as they do not compute input dependent attention weights and have weaker representation power. Our goal in studying them is to see if they can substitute some of the expensive attention layers for computing the contextual mappings. These models are trained in a large batch setting, with a batch size of 8192 for 60k steps, unlike the other set of experiments reported in Fig. 1. Note that average attention has clearly worse performance, showing that theses tasks indeed require an advanced architecture.

## D  EXPERIMENTAL SETUP

For our experiments we follow the same setting as in BERT (Devlin et al., 2018). We first pre-train the models on the masked language modeling task and the next sentence prediction task. We use English Wikipedia corpus and BooksCorpus dataset (Zhu et al., 2015) for this pre-training. We use BERT$_{\text{BASE}}$, a 12 layer Transformer model as the baseline. This model uses an embedding size of 768 and has 12 head self-attention layers and 3072 wide feed forward layers. We train it with the Adam optimizer, with .01 dropout and weight decay. We do pre-training for 250k steps with a batch size of 1024 and a max sequence length of 512. Pre-training takes around 2 days on 16 TPUv3 chips. We take the pre-train models and finetune them on the MNLI and SQuAD datasets separately using the same hyper-parameters as in Devlin et al. (2018). MNLI is a sentence entailment task in which, given a premise sentence, requires us to classify a hypothesis sentence into neutral, contradiction or entailment classes. We report the classification accuracy on this task. SQuAD is a question answering task, in which given a paragraph and a question, requires us to identify the answer as a span of the words in the paragraph. For this task we report both the F1 score and the Exact Match (EM) percentage. The metrics are reported on the dev sets of these datasets.

For our experiments with the depth-wise separable convolution layers, we follow the implementation in (Wu et al., 2019). We first use a GLU layer followed by the convolution layer. We use 16 separable convolution filters, of filter length 128, and reuse them, with each filter operating on 48 of the 768 dimensions of the input. This layer also has a skip connection and the output is normalized using layer normalization, similar to the self-attention layer. In our experiments, we replace the self-attention layers of the Transformers, in the lower layers, with this convolution layer. We keep the feed forward layer of the Transformer block the same.

For the experiments performed in this paper, one might consider an alternate explanation that the tasks considered maybe are easy, and do not require any advanced architecture to solve them, and even a simple architecture (bi-linear projection or separable convolution) might solve these tasks. To rule out this case we consider an even simpler architecture, namely average attention, as a baseline for our experiments.

**Average attention.** An average attention layer replaces the self-attention layer, and just computes the average of projections of all the other tokens. That is, we replace $\sigma[(\boldsymbol{W}_K^i \boldsymbol{X})^T \boldsymbol{W}_Q^i \boldsymbol{X}]$ in (1) with a matrix full of $1/n$. The model still has the skip connections and the feed-forward layers like Transformer.

