# OpenReview forum: "Are Transformers universal approximators of sequence-to-sequence functions?"
_ICLR.cc/2020/Conference — Accept (Poster)_

### Official Review · AnonReviewer1 · 2019-10-21
**Official Blind Review #1**

**Rating:** 6

**Review:**


This paper discusses the universal approximation capability of the Transformer, under certain assumptions, analyze the role of different components of the Transformer (e.g., self-attention layer for contextual mapping), and propose the use of some other layers that can also provide contextual mapping.

Overall speaking, the problem studied by this paper is very important. The transformer has been used extensively in many applications today, however, deep theoretical understanding of it is not sufficient. Universal approximation capability is a very important theoretical property of deep learning, and advances on the universal approximation of the Transformer is important for the deep learning community. Therefore, I think people will be willing to see the results in this paper.

While saying so, this paper has some limitations, which could be further improved.

1)	The paper studies a variant of the Transformer, where the layer norm is removed. However, according to practical experiences, the layer norm plays a critical role in the Transformer. As a result, there is gap between this paper and practical situations, and the value of the paper becomes not very clear.

2)	The paper lacks experimental verifications. It would be better to design some toy experiments with different types of target functions to see whether the Transformer can well approximate them, and see the contribution of different components of the Transformer

3)	The discussions on contextual mapping are not solid enough. First, it seems that contextual mapping is kind of sufficient condition for the proof, however, it is unclear whether it is a necessary condition. Consequently, things are not clear regarding”

a.	If all the structures (self-attention, bilinear projection, depth-wise separable convolutions) are all sufficient conditions, why self-attention is better than the other two, and why the mix of them can generate even better results?

b.	If they are not necessary conditions, we cannot say one should choose them since other structures may be equally good even if they do not satisfy contextual mapping conditions.

4)	The experimental study in the paper is not very comprehensive. Given that the Transform has been used in many NLP scenarios, experiments on more datasets and more tasks are expected.


**I read the author rebuttal. Some of my concerns still remain, and I would like to keep the current rating (which is positive).

**Experience Assessment:**

I have published in this field for several years.

**Review Assessment: Checking Correctness Of Derivations And Theory:**

I assessed the sensibility of the derivations and theory.

**Review Assessment: Checking Correctness Of Experiments:**

I carefully checked the experiments.

**Review Assessment: Thoroughness In Paper Reading:**

I read the paper at least twice and used my best judgement in assessing the paper.

---

> ### Author Response · Authors · 2019-11-09
> **Author Response to Review #1**
>
> We thank the reviewer for their review. We answer the points raised by the reviewer, listed by the item number:
>
> 1) Even though it is true that our model does not have layer normalization, note that generally the purpose of normalization techniques, such as batch norm or layer norm, is to improve the optimization dynamics, not necessarily the expressive power of the model. To make the analysis simpler, we decided to study a model without normalization layers. In fact, many theoretical papers on neural networks (e.g., universal approximation [1, 2, 3], optimization dynamics [4, 5, 6], and many others) also omit normalization layers in their analysis, despite the fact that normalization layers such as BatchNorm are commonly used in practice.
>
> 2) We’d like to emphasize that the focus of our paper is on theoretical analysis of Transformers. Universal approximation property requires exponential depth or width to approximate arbitrary functions, which is hard to verify empirically. We decided instead to focus our experimental efforts on evaluating simpler alternative structures to the attention layers.
>
> 3) It is true that contextual mapping is a “sufficient” condition for our proof because it is specific to our proof strategy.
>
> 3a) Bilinear projection and depth-wise separable convolution evaluated in our experiments can implement contextual mapping *to some extent*, and they do not implement the full contextual mapping as proved in Lemma 6. We will make this clearer in the updated version. We do not expect these cheaper models to have the same performance as attention, because they do not use input-dependent weights (as done in attention) to compute the output embeddings. The purpose of our experiments is only to see if we can substitute some of the expensive attention layers with these cheaper layers.
> Our experiments show that attention layers indeed perform better than the other two layers. However, we observed that replacing the first two layers can surprisingly help performance; the best explanation we have is that the first few attention layers tend to attend broadly to the whole sequence (as empirically observed in [7]), so the alternative layers can perform the job more efficiently.
>
> 3b) Although we don’t formally prove that the capability to implement contextual mappings is a necessity in our paper, it is necessary for universal approximation because
> i) token-wise feedforward layers cannot capture interaction between tokens, so capturing “context” should be done by attention layers (or other alternatives to attention), and
> ii) in order to implement arbitrary mappings, attention layers need to be able to distinguish between different contexts.
>
> 4) We’d like to again note that the primary focus of our paper is to theoretically analyze the expressive power of the Transformers. As the reviewer mentioned, Transformer has been successfully used in many NLP scenarios, which motivated us to theoretically investigate its expressive power and prove our universal approximation theorem. The purpose of our experiments is only to provide some preliminary insights into substituting some of the expensive attention layers with simpler architectures, and a detailed evaluation of such hybrid architectures will be interesting future research.
>
> [1] The Expressive Power of Neural Networks: A View from the Width. https://arxiv.org/abs/1709.02540
> [2] Approximating Continuous Functions by ReLU Nets of Minimal Width. https://arxiv.org/abs/1710.11278
> [3] Universal Approximation with Deep Narrow Networks. https://openreview.net/forum?id=B1xGGTEtDH
> [4] Gradient descent provably optimizes over-parameterized neural networks. https://arxiv.org/abs/1810.02054
> [5] A Convergence Theory for Deep Learning via Over-Parameterization. https://arxiv.org/abs/1811.03962
> [6] Stochastic Gradient Descent Optimizes Over-parameterized Deep ReLU Networks. https://arxiv.org/abs/1811.08888
> [7] What Does BERT Look At? An Analysis of BERT’s Attention. https://arxiv.org/abs/1906.04341

---

### Official Review · AnonReviewer2 · 2019-10-24
**Official Blind Review #2**

**Rating:** 6

**Review:**

CONTRIBUTIONS:

C1. Transformers (without positional encodings and without layer normalization), with 2 attention heads of dimension 1 and feed-forward layers (FFN) with 4 hidden nodes, are universal approximators of continuous permutation-equivariant functions f of compact support, relative to any Lp metric (1 <= p < \infty). (Thm. 2, p. 3). (Without positional encodings, a function f computed by a Transformer is permutation equivariant: f(P(X)) = P(f(X)) for any permutation P of the columns of X, which are the vector encodings of the input tokens.)

C2. For transformers with trainable positional encodings, the same result holds without the restriction to permutation equivariance (Thm. 3, p. 4)

C3. The notion of ‘contextual mapping’ is introduced (Def. 5.1, p. 6); this notion is central to the proofs of the theorems in C1-C2. Such a mapping q takes any two vectors L, L’ in a finite subset of a real vector space and maps them to a vector space such that all elements (in R) of q(L) and q(L’) are distinct. (In the permutation-equivariant context, L’ must not be a permutation of L.)

RATING: Weak accept

REASONS FOR RATING (SUMMARY). ‘Accept’ because the results seem important. ‘Weak’ because the explanation for the crux of the key result is inaccessible. It relies crucially on the notion of C3, the utility of which for understanding how real Transformers work is questionable.

REVIEW

Sec. 4 does a good job of sketching the key proof. But Sec. 5 falls down at a crucial point, Lemma 6.


What I can glean from the paper (Sec 5.0) is that the idea of the proof is this: (1) use the continuity of f to approximate it with a piecewise constant function c, which takes a fixed value within each hypercube of a discretization of the compact support of f; (2) use a FFN g to map f’s input space to a discrete subset, a regular finite grid G defining the corners of the hypercube discretization; (3) use modified Transformer attention layers to create a contextual mapping k of G; (4) use a FFN h on R to map each (unique) real number in the vector outputs of k to the correct output value so that h([k(g(X))]_i) = [c(X)]_i; (5) approximate the modified Transformer attention layers of k with standard Transformer attention layers. (The modified attention layers replace softmax with argmax and replace ReLU with a (varying) piecewise-linear activation functions “with at most 3 pieces, at least one of which is constant” [Step 2, p. 4.])

The paper should focus on steps (3) and (5), the only parts of the proof that pertain to what is special about the Transformer: attention. These should be explained fully and clearly in the main text. To make room for this, the rest, concerning FFN approximation and discretization, can be reduced to a paragraph each in the main text, since they are standard, not particular to Transformers.

My attempts to understand the proofs of (3) and (5) from the Appendix were not successful. To illustrate the kind of difficulty I had in several places: in the proof of Lemma 9 on p. 12, the conclusion is “Thus, given any \bar{g} \in \bar{\cal{T}}^{2,1,1}, there exists a function g \in \cal{T}^{2,1,4} arbitrarily close to \bar{g}, by appropriately choosing the parameters to be large enough.” (“2,1,4” means a Transformer attention layer with 2 heads of dimension 1 followed by a FFN with 4 hidden units) But how do the numbers {2,1,1} and {2,1,4} relate to the equations internal to this proof? I can see some possible connections, but the authors should spell this out very clearly rather than making the reader work to fill in the gap. It should be clear how the ATTENTION MECHANISM is crucial for the proof by understanding just how that mechanism does the work needed in steps (3) and (5) above.

Given the time-consuming but ultimately unsuccessful attempt to understand the first few pages of the 11-page Appendix, I did not attempt to absorb the remaining pages.

It is clear that the notion of contextual mapping plays an important role in their proof of the universal approximation theorem, but does it play any role in the operation of Transformers in practice? Spraying the points of a grid G in the Transformer’s input space into a collection of vectors in which the same real number never appears twice is a nice step in an approximation proof, because it reduces the problem to mapping a set of unique real numbers to another set of real numbers (the elements of the output vectors). But does an actual trained Transformer in practice do anything resembling this? It seems implausible on the face of it, but perhaps there is a theoretical argument, or empirical evidence, that makes it plausible. Thus, for understanding how real Transformers actually do their work, the value of the notion of contextual mapping, and therefore the discussion at the end of the paper of alternatives to attention for achieving it, appears questionable to me. The subtitle of Sec. 5, “Demystifying Transformers”, is not clearly justified.

**Experience Assessment:**

I do not know much about this area.

**Review Assessment: Checking Correctness Of Derivations And Theory:**

I assessed the sensibility of the derivations and theory.

**Review Assessment: Checking Correctness Of Experiments:**

I did not assess the experiments.

**Review Assessment: Thoroughness In Paper Reading:**

I read the paper at least twice and used my best judgement in assessing the paper.

---

> ### Author Response · Authors · 2019-11-09
> **Author Response to Review #2**
>
> We appreciate the reviewer’s time and thoughtful comments. Below, we provide answers to the reviewer’s concerns, by the order they appear.
>
> “What I can glean from the paper (Sec 5.0) is that the idea of the proof is this…”
> - Your understanding is correct!
>
> “The paper should focus on steps (3) and (5)...”
> - We agree that the main text does not provide enough intuition on the proofs of lemmas involving attention layers, especially Lemma 6. In our initial submission, we had decided not to include proof sketch of Lemma 6 in the main text because its proof is technically involved and takes up 5 pages in the appendix. Conceding to your point, we agree that we need to have a brief proof sketch of Lemma 6 in the main text, considering that Lemma 6 is actually the main technical contribution of this paper. We will update the paper shortly after this response.
>
> “But how do the numbers {2,1,1} and {2,1,4} relate to the equations internal to this proof?”
> - Please recall that T^{2,1,4} means Transformers with 2 attention heads with head size 1 in the attention layer and 4 hidden nodes in the feed-forward layer.
> We showed in the proof that using large enough $\lambda$, softmax (\sigma) can approximate hardmax (\sigma_H). This means that 2 attention heads with head size 1 in *modified* attention layer can be approximated with *standard* attention layer of the same size, by choosing large enough parameters.
> Also, we showed that any piecewise activation function $\phi$ in $\Phi$ can be approximated with the sum (\tilde \phi) of 4 ReLUs arbitrarily closely when $\epsilon$ goes to zero. This means that 1 hidden node in the *modified* feed-forward layer can be approximated with 4 hidden nodes in the *standard* feed-forward layer. This is why we can approximate any modified Transformer-(2,1,1) with a standard Transformer-(2,1,4).
>
> “It should be clear how the ATTENTION MECHANISM is crucial for the proof ...”
> - In fact, the proof of Lemma 9 (i.e., step (5)) does not have anything specific to do with attention layers, other than approximating hardmax with softmax. For Lemma 6 (i.e., step (3)), attention layers are indeed crucial in the construction. As noted earlier, for Lemma 6, we will update the manuscript with more details of the proof in the main text.
>
> “It is clear that the notion of contextual mapping … but does it play any role in the operation of Transformers in practice?”
> - Since Transformers used in practice have fixed depth, we believe that Transformers in practice might not be able to exactly implement contextual mappings as we defined in our paper. However, there is some preliminary empirical evidence that Transformers do implement some sort of “contextual mappings,” e.g., as reported in [1]. Figure 4 of [1] shows how the word “die” is mapped to different points in the embedding space, depending on the contexts in which they appear. Detailed analysis of the nature of embeddings learned by Transformers is an interesting direction for future research.
>
> [1] Visualizing and Measuring the Geometry of BERT. https://arxiv.org/abs/1906.02715

---

> > ### Comment · AnonReviewer2 · 2019-11-15
> > **AnonReviewer2 Response**
> >
> > Thank you to the authors for their responses. The addition to the main text of a more intuitive explanation of Lemma 6 is much appreciated. I do not find particularly convincing the response to the question of whether computing ‘contextual mappings’ is an insight into the actual workings of actual Transformer models, since the fact that the same word yields different representations in different contexts is hardly an insight for which this intense formal analysis is needed to appreciate. What I question is the relevance in practice of the particular definition of ‘contextual mapping’ that requires all the real numbers comprising all the elements of a set of representational vectors to be distinct, a very delicate requirement which I would be astounded to find reflected in the operation of actual Transformers. However, the main contribution of the paper is in the realm of function approximation theory, not the realm of understanding actual Transformer models, so I do not think this weakness detracts significantly from the paper’s importance. I am therefore raising my score to 8: Accept.

---

### Official Review · AnonReviewer4 · 2019-11-02
**Official Blind Review #4**

**Rating:** 6

**Review:**

This paper tries to analyse the Transformer, widely applied building block of a neural network component, to improve understanding of the internals of the model. The analysis starts showing that the transformer blocks generate permutation equivalent maps and then shows that the transformer can approximate any permutation equivalent map in a compact domain with arbitrary precision. Three key steps are developed and used to prove the universal approximation of arbitrary permutation equivalent map: 1) quantization of input via feed-forward layers, 2) contextual mapping via attention layers, and 3) value mapping via feed-forward layers. By introducing positional embeddings, the paper relaxes the restriction on permutation equivalence and proves that the Transformer is a universal approximator of any sequence to sequence function.

Overall, the paper presents an interesting analysis of the Transformer providing some practical implications, with a caveat for some clarifications on the experiments. The structure of the manuscript could be improved.

One of the natural question after reading this paper is whether the claims made in section 5 are what actually happens inside of the Transformer because we often observe the attention layers of the first few stacks of Transformer blocks do something. Since the claims lead us to have a universal approximator of seq-2-seq functions, it would be great if there is an experiment based on an alternative model structure based on the claims. For example, one can design a new Transformer architecture with stacks of feed-forward layers, followed by stacks of self-attention layer, followed by other stacks of feed-forward layers. Which may reduce the number of model parameters while preserving a similar level of performance.

Here are a few minor comments on the structure of the manuscript. It seems a bit unnatural to have a section with a proof sketch (section 4). Will it be better if it follows theorem 2. Also, the title of section 5 seems not very informative (proof sketch of proposition 4). You may consider rewriting these sections to improve readability. The definition of contextual mapping and the following lemma seem also one of the major contribution of this paper since the other proof techniques are somewhat familiar with the existing method on universal approximation theory. It would be good to have these result in the early part of the manuscript instead of having it in the end.

**Experience Assessment:**

I do not know much about this area.

**Review Assessment: Checking Correctness Of Derivations And Theory:**

I did not assess the derivations or theory.

**Review Assessment: Checking Correctness Of Experiments:**

I assessed the sensibility of the experiments.

**Review Assessment: Thoroughness In Paper Reading:**

I read the paper at least twice and used my best judgement in assessing the paper.

---

> ### Author Response · Authors · 2019-11-09
> **Author Response to Review #4**
>
> Thank you for your efforts in reviewing our paper. Below, we address the concerns raised.
>
> “... whether the claims made in section 5 are what actually happens inside of the Transformer…”
> - We note that our universal approximation theorem is based on a constructive proof, so it does not necessarily describe what Transformers might actually be learning in practice. It shows that they have the ability to approximate arbitrary functions. Having said that, we would like to emphasize that preliminary empirical observations show that Transformers do implement some sort of “contextual mappings”, e.g., see Figure 4 in [1]. The figure shows how the word “die” is mapped by a Transformer to different points in the embedding space, depending on different contexts in which they appear.
>
> - Designing a new Transformer architecture is indeed an interesting direction to pursue. In fact, we have found some recent works attempting to reorder the attention and feedforward layers of the Transformer [2]. However, please note that better performance in practice requires, in addition to good expressive power, also good optimization dynamics, studying which is another interesting research direction to pursue.
>
> “few minor comments on the structure of the manuscript”
> - The definition of contextual mapping and Lemma 6 is indeed one of the key technical components of our theorems. We will shortly update the paper according to the comments.
>
> [1] Visualizing and Measuring the Geometry of BERT. https://arxiv.org/abs/1906.02715
> [2] Understanding and Improving Transformer From a Multi-Particle Dynamic System Point of View. https://arxiv.org/abs/1906.02762

---

### Author Response · Authors · 2019-11-14
**Summary of revision**

We have updated the paper according to the reviews and our response.

To list some of the changes:
- Following Reviewer 4's comment, we restructured the paper to bring the definition of contextual mappings and a sketch of Lemma 6 to an earlier part of the paper.
- Following Reviewer 2's comment, we added a sketch of proof of Lemma 6 to the main text (Section 4.2.1).
- In accordance with our response to Review 1, we edited the experimental section to make it clearer that BProj and SepConv do not implement our contextual mappings (Definition 3.1). We also edited the discussion on the experiments to reflect reviewers' comments and our response.
- In accordance with our response to Review 2, we added more details to the Proof of Lemma 9.

We would appreciate it if the reviewers could take a look at our revised paper. Thank you!

Best regards,
Paper1020 Authors

---

### Decision · Program_Chairs · 2019-12-19

**Decision:**

Accept (Poster)

**Comment:**

The paper provides a proof that Transformer networks (a popular deep learning model) are universal approximators for sequence-to-sequence functions. The theorem relies on the idea of contextual mappings (Definition 3.1), which models the attention layers. The results provide an important starting point for understanding a very widely used architecture.

As with many theoretical papers, the reviewers provided several suggestions as to which are important parts to be presented in the main paper. The authors were very responsive during the discussion period, updating the structure of the paper significantly. This shows nice evidence supporting the need for a long discussion period for ICLR. One reviewer upgraded their score (to 8), which is not reflected in the system.

This is an excellent paper, providing much needed theoretical analysis of a popular neural architecture. Clear accept.